# Heparin-network-mediated long-lasting coatings on intravascular catheters for adaptive antithrombosis and antibacterial infection

Lin Liu[1,2,4], Huan Yu[1,2,4], Lei Wang [1], Dongfang Zhou [3], Xiaozheng Duan [1], Xu Zhang [1], Jinghua Yin[1], Shifang Luan [1,2] ✉ & Hengchong Shi [1,2] ✉

Bacteria-associated infections and thrombosis, particularly catheter-related bloodstream infections and catheter-related thrombosis, are life-threatening complications. Herein, we utilize a concise assembly of heparin sodium with organosilicon quaternary ammonium surfactant to fabricate a multifunctional coating complex. In contrast to conventional one-time coatings, the complex attaches to medical devices with arbitrary shapes and compositions through a facile dipping process and further forms robust coatings to treat catheter-related bloodstream infections and thrombosis simultaneously. Through their robustness and adaptively dissociation, coatings not only exhibit good stability under extreme conditions but also significantly reduce thrombus adhesion by 60%, and shows broad-spectrum antibacterial activity ( > 97%) in vitro and in vivo. Furthermore, an ex vivo rabbit model verifies that the coated catheter has the potential to prevent catheter-related bacteremia during implantation. This substrate-independent and portable long-lasting multifunctional coating can be employed to meet the increasing clinical demands for combating catheter-related bloodstream infections and thrombosis.

Intravascular (IV) catheters are widely used in the clinic to diagnose and treat acute and chronic diseases, especially for patients with heart disease, cardiovascular disease, cancer, and so on[1]. Approximately 90% of patients admitted in hospital require short- or long-term IV catheters to deliver medications, nutrient solutions, and blood[2]. More than 5 million central venous catheters (CVCs) are applied to treat cancer in the USA[3,4]. However, the widespread use of IV catheters may largely increase the occurrence of life-threatening complications, such as catheter-related bloodstream infections (CRBIs)[5] and catheter-related thrombosis (CRT)[6]. Blood clots on the surfaces of catheters may also lead to device dysfunction and block blood circulation, resulting in

deep vein thrombosis with high mortality[7]. For instance, deep vein thrombosis associated with CRT accounts for 10% of CRTs in adults and 50–80% in children[6]. In addition to CRT and CRBIs, bacteria introduced during the implantation of IV catheters could lead to bacteremia-caused disease due to the formation of biofilms[8], such as endocarditis[9]. Although great efforts have been devoted to preventing bacterial infection[10–12], the mortality rate of CRBIs remains high at 12–25% annually in the United States[4,13]. Importantly, CRBIs and CRT are complex and interlinked[14,15]. Bacterial toxins can cause clots to form[14,16], and thrombi can easily result in bacterial adherence to enhance the formation of biofilms due to their components, such as

[1]State Key Laboratory of Polymer Physics and Chemistry, Changchun Institute of Applied Chemistry, Chinese Academy of Sciences, Changchun 130022, China. [2]University of Science and Technology of China, Hefei 230026, China. [3]NMPA Key Laboratory for Research and Evaluation of Drug Metabolism, Guangdong Provincial Key Laboratory of New Drug Screening, School of Pharmaceutical Sciences, Southern Medical University, Guangzhou 510515, China. [4]These authors contributed equally: Lin Liu, Huan Yu. ✉e-mail: sfluan@ciac.ac.cn; shihc@ciac.ac.cn

fibrin and fibronectin[17]. Bacterial colonization rates on catheters with thrombosis were almost double (32% vs. 19.4%), and the catheter sepsis rates were increased (19% vs. 7%)[18]. Despite significant efforts[1], the interrelationship between infectious and thrombotic complications remains unclear, causing serious clinical challenges, and an effective approach to treat CRBIs and CRT simultaneously is urgently needed.

Normally, a common strategy to treat infection and thrombi in the clinic is to administer antibiotics and intravenous heparin (HS) prior to or immediately following catheter implantations. However, the generation of antibiotic-resistant bacteria and heparin-caused thrombocytopenia are still challenges for these treatments[19]. Thus, constructing functional IV catheter devices with antibacterial and antithrombin effects may be a more appealing approach to decrease CRT or CRBIs. Antibacterial catheters can be obtained by chemically modifying cationic polypeptides[20,21], quaternary ammonium compounds and other cationic moieties[22]. Thrombi can be reduced by coating with heparin, corn trypsin inhibitor, and thrombomodulin[23–26]. However, commercially available approaches to treat both CRBIs and CRT are lacking due to the complexity and interrelation of CRBIs and CRT. More importantly, the potential interrelationship between CRBIs and CRT is usually ignored[9]. Additionally, the diversity of catheters with various lumens and narrow and long bores could further increase the difficulty of forming coatings. Covalent grafting, which is used to construct robust coatings, usually involves a rigorous process. In contrast, noncovalent coatings with rapid and facile features may be limited by the poor stability under complex physiological environments, including variations in ionic strength, pH, temperature, etc[27–30]. Hence, a robust (i.e., displaying long-term stability for up to 30 days in diverse environments), modular (structural and compositional flexibility), and adaptable (stimulus-responsive) coatings should be developed to obtain functional catheters with both antibacterial and antithrombotic properties to simultaneously treat CRBIs and CRT.

Herein, we report a concise and straightforward route to construct a long-lasting (30 days) and adaptable coating that combines the high antithrombin ability of HS and good antibacterial feature of dimethyloctadecyl[3-(trimethoxysilyl)propyl]ammonium (DAC) to synergistically treat CRBIs and CRT. As depicted in Fig. 1, polyanionic HS-Na can coassemble with DAC-Cl to generate HS/DAC aggregates via a one-step reaction. Due to its high adhesiveness, the liposoluble HS/DAC coating can adsorb tightly on substrates, exhibiting great promise for use in various IV catheters, such as metre-long and thin catheters. After heating, the formation of cross-linked networks between HS/DAC coatings and substrates could further enhance the structural stability even under various extreme environments and mechanical bending. Meanwhile, salts could weaken the electrostatic dipole-dipole interactions among intrachain of HS/DAC. More functional groups in HS/DAC, such as sulfamate, sulfonate and carboxyl groups of HS and quaternary ammonium groups of DAC, are exposed in electrolyte solution. The exposed hydrophilic groups are more conducive to reducing the thrombus adhesion rate (84.6%) in acute canine models and also exhibit a broad-spectrum antibacterial performance against *Staphylococcus aureus* (*S. aureus*) and *Escherichia coli* (*E. coli*). Moreover, we also verified that this coating could largely decrease bacterial infections and thrombus caused by catheter implantation in an ex vivo rabbit model, which has rarely been reported.

## Results

### Formation of stable HS/DAC coating

We first prepared the HS/DAC complexes via a concise and direct self-assembly process (Supplementary Fig. 1). With increasing alkyl chain length of DAC from 1 to 18, a lower concentration was needed to generate water-insoluble aggregates (Supplementary Fig. 2). Owing to the strong hydrophobic interactions, DAC with alkyl chain length of 18 could generate more water aggregates than the other groups

(alkyl chain < 18) at the same concentration of $4 \times 10^{-5}$ mM, and this HS/DAC complex was chosen for the following characterizations. Compared with HS, the HS/DAC complex generated new infrared characteristic bands (such as the -$CH_3$ absorption of DAC at 2853 cm$^{-1}$) and new peaks in X-ray photoelectron spectra (XPS, such as Si 2$p$ of DAC at 101 eV), indicating that the HS/DAC complex was successfully synthesized (Supplementary Fig. 3). Good solubility in organic solution should also be considered when constructing uniform coatings, particularly for IV catheters with complicated compositions and shapes. As shown in Supplementary Fig. 4, the liposoluble HS/DAC complex can form a clear solution in dimethyl sulfoxide, dichloromethane, and toluene, which is ascribed to the long alkyl groups and electrostatic screening effect.

Given its water-insoluble yet liposoluble characteristic, the HS/DAC complex serves as an optimal candidate for rapidly generating uniform coatings via a facile dip coating method. As shown in Supplementary Fig. 5a, the HS/DAC complex initially attached to the hydroxylated thermoplastic polyurethane (TPU) surfaces via intermolecular interactions, and a robust coating could be obtained by a heat treatment to form chemical cross networks among the hydrolysable alkoxysilane of HS/DAC and the hydroxy groups in substrates, DAC, or HS. Water contact angles, XPS, and Fourier Transform Infrared (FTIR) were used to confirm the formation of HS/DAC on the TPU surfaces (Fig. 2a and Supplementary Fig. 5b, d)[31]. After a heat treatment, a new band at 1118 cm$^{-1}$ appeared in the FTIR spectrum of the (HS/DAC)-TPU, belonging to an asymmetrical telescopic vibration of Si-O-Si. Obvious changes in absorption intensity at 1080 cm$^{-1}$ and 932 cm$^{-1}$ were attributed to the formation of Si-O associated bands, including Si-O-Si, Si-OH, or Si-O-C[32–36]. Compared with the bare TPU, no noticeable changes in roughness were observed in the formed HS/DAC coating (Fig. 2b), and its thickness was approximately 170 nm (Fig. 2c). Additionally, more than 80% transmittance was maintained in HS/DAC-coated TPU, suggesting that the formed coating exhibits good optical transparency and is an ideal coating material for medical devices (Fig. 2d).

Due to the complex shapes and clinical environments of applied medical devices, coating materials must involve facile manufacturing methods and exhibit long-term robustness. To demonstrate this, the HS/DAC complex was utilized to form coatings on various medical devices regardless of the composition and shape via the staining method (Fig. 2e), including polyurethane multi-lumen CVCs, polyvinyl chloride (PVC) hemodialysis catheters, glass and slender catheters (metre-long, outside diameter < 1 mm). Next, the interfacial bonding strength of the coating under different extreme environments was evaluated by using fluorescent probes. A common quaternary ammonium salt without a siloxanyl functional group (stearyl trimethyl ammonium chloride, QAC-Cl) was used to prepare the HS/QAC complex as a control. The chemical stabilities of (HS/DAC)-TPU and (HS/QAC)-TPU control under simulated environments, including salts, organic solvents, and pH values, were monitored by the colour or fluorescent changes of Rose Bengal and fluorescein sodium detectors, respectively. After 3 h of ultrasound in the simulated solution, the noncovalently fixed HS/QAC coating detached significantly, especially in high-concentration salt solutions and disinfectants. In contrast, the (HS/DAC)-TPU with strong chemical cross-linking bonds exhibited uniform and changeless colour or fluorescent distributions under all tested conditions (Fig. 2f i and Supplementary Fig. 6). Additionally, scanning electron microscopy (SEM) images were obtained to visualize the morphology changes in the coating under extreme challenges, confirming the mechanical stability of HS/DAC coating. After bending to 180° for 500 times, HS/DAC-coated catheter showed little delamination or cracking, but large amounts of delamination and cracks were present on the surface of the HS/QAC-coated catheter control (Fig. 2f ii). After (HS/DAC)-TPU underwent autoclaving (120 °C, 20 min) or was exposed to liquid nitrogen for 30 min, no peeling was observed and

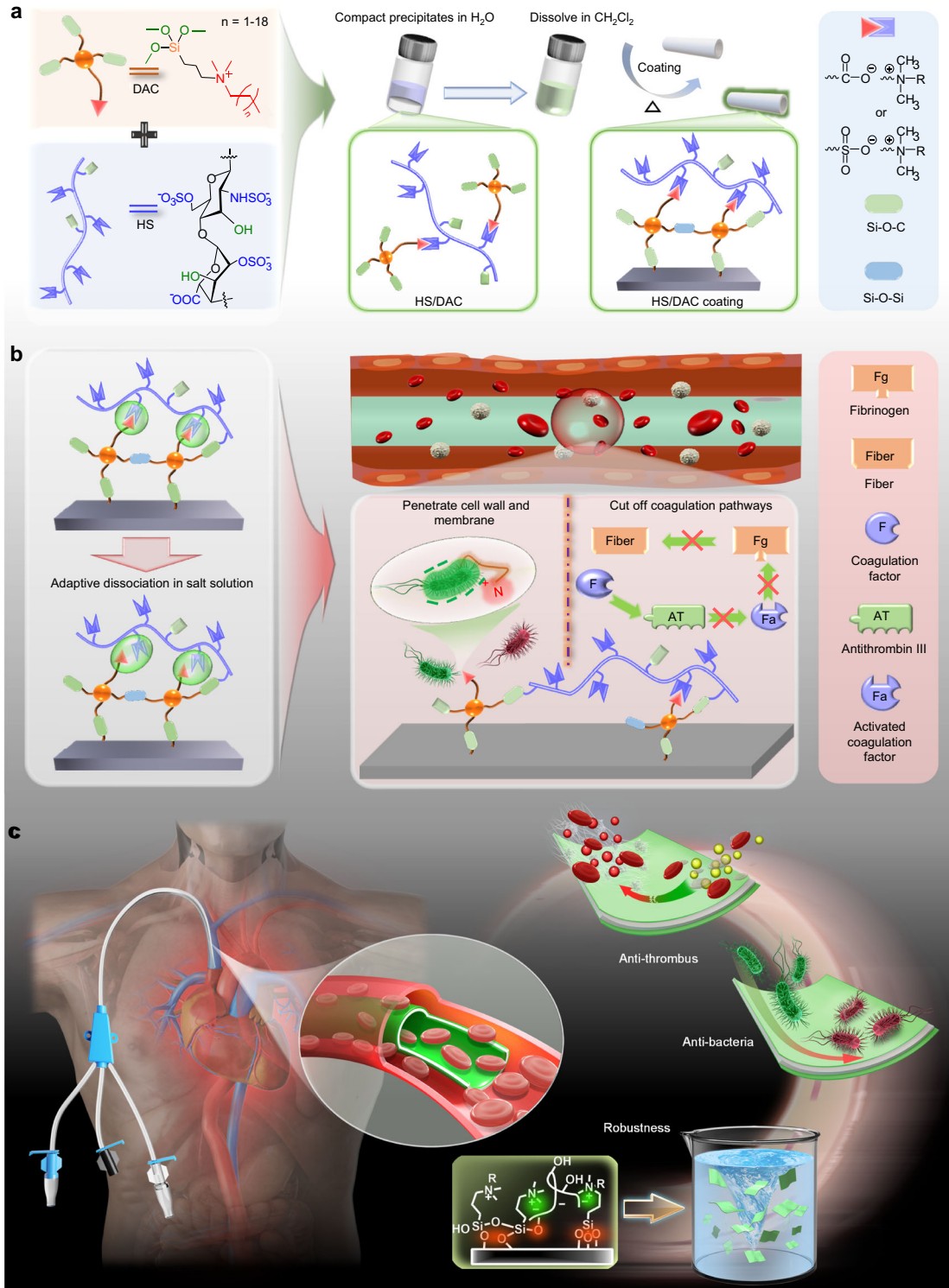

**Fig. 1 | Schematic illustration showing the formation and application of HS/DAC-coated CVCs with robust, antibacterial and antithrombotic properties.** **a** Illustration of the procedure used to prepare the HS/DAC complex and HS/DAC coating. **b** Dissociation properties of HS/DAC coatings and the resulting anti-bacterial and antithrombotic capabilities. **c** Multifunctionality of HS/DAC-coated IV catheters, such as robustness and antibacterial and antithrombotic capabilities.

the coating was intact (Fig. 2f iii). We further explored the long-term stability of HS/DAC coating in flowing 0.9% NaCl via a multichannel circulation model and in vivo stability via a subcutaneous implantation model, respectively. Compared with the untreated HS/DAC coating, the absorbance of the labeled coating decreased by 1.8 times and 3 times after flow incubation and implantation in vivo for 30 days (Supplementary Fig. 7).

## Anti-thrombotic property of the HS/DAC coating

The formation of CRT is a multi-component pathology after catheter implantation into blood vessels, such as various coagulation factors, proteins, blood cells, and inflammatory cells, etc[37,38]. Therefore, the anticoagulant mechanism of the coating was firstly investigated by exploring the relationship between the surface and the coagulation cascade, protein, and inflammatory cells.

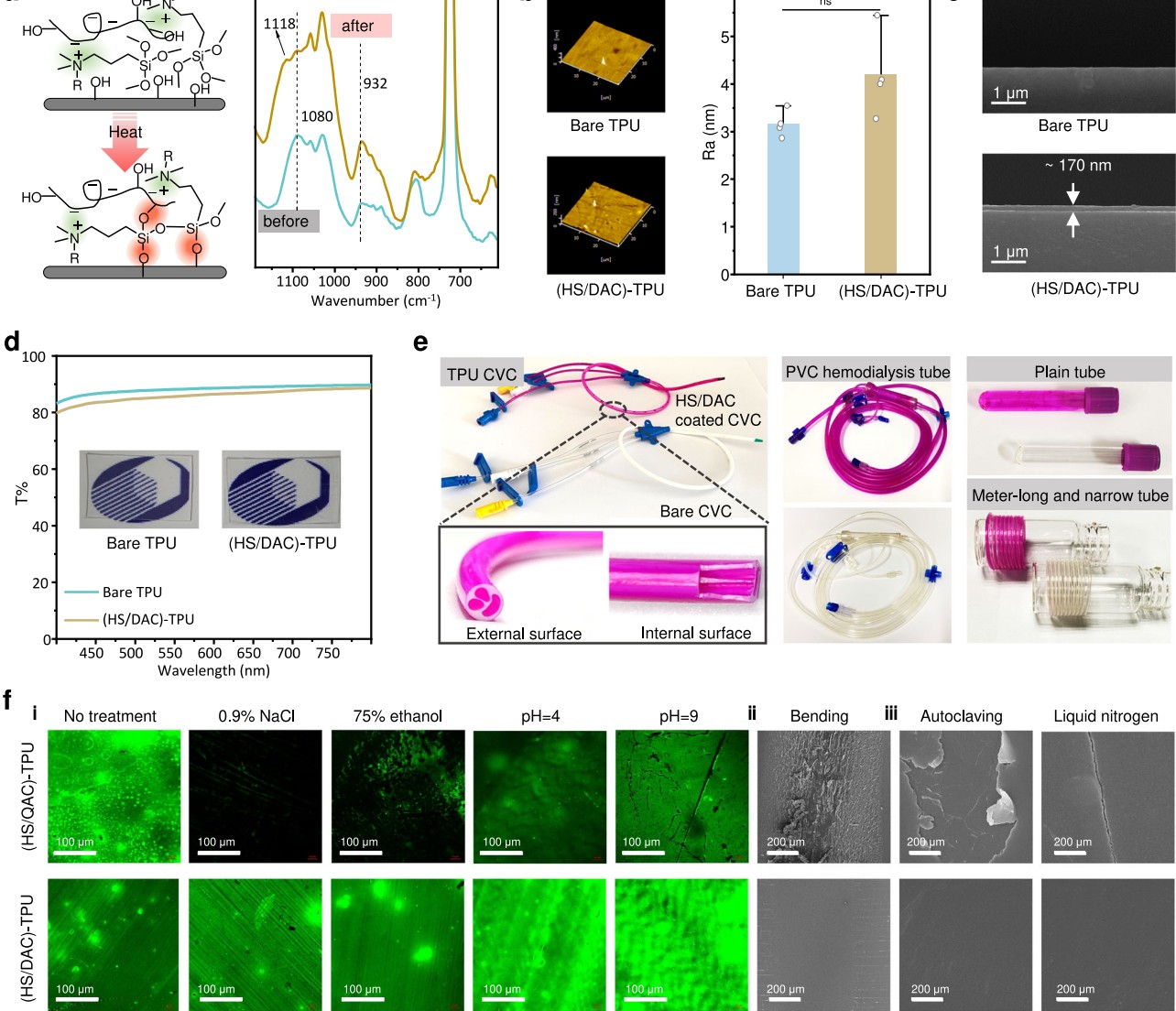

**Fig. 2 | Preparation and characterization of the HS/DAC coating. a** Schematic illustration of the condensation process of the HS/DAC coating after heating and the corresponding changes in Si-O associated bands in the FTIR. **b** The roughness of bare TPU and HS/DAC-coated TPU film via AFM (mean ± SD, $n = 4$ independent samples). **c** SEM images of cross sections of bare TPU and HS/DAC-coated TPU films. Measurements were repeated four times independently with similar results. **d** Transmittance of bare and HS/DAC-coated TPU film. **e** Photographical images of the HS/DAC coating on different medical devices. **f** Robustness of the HS/DAC coating under (**i**) organic reagents, electrolyte solutions and solutions with different pH values, (**ii**) bending and (**iii**) extreme environments. Measurements were repeated three times independently with similar results. Two-sample Student's t test was used in b, ns: no significant difference, $P = 0.071$.

As previously demonstrated[39,40], HS catalyzes the activity of the serpin antithrombin III and forms hydration layers to obtain antithrombin activity. The conformation of antithrombin III changes after binding to HS, which can greatly inhibit the coagulation factor Xa (FXa) and thrombin (FIIa). To examine the anticoagulant activity of the HS/DAC coating, chromogenic anti-FXa and anti-FIIa assays were used. Compared with TPU and DAC-TPU, (HS/DAC)-TPU reduced the substrate conversion by 95.8% and 94.3%, respectively (Fig. 3a, b), indicating that (HS/DAC)-TPU significantly inhibited FXa and FIIa. Therefore, the HS/DAC coating may inhibit the formation of thrombi by cutting off some coagulation pathways.

Fibrinogen (Fg) activation to fibrin is the final step of the coagulation cascade, in which the hydrolysis of thrombin plays a crucial role. HS/DAC coating has been proven to have a high inhibitory effect on thrombin activity. Then, the adhesion/activation of Fg were further investigated through a closed circular tube loop of whole blood (Supplementary Fig. 8a). FITC-Fg was added to fresh blood and cycled

with samples for 2 h to study Fg adhesion. Confocal laser scanning microscopy (CLSM) images indicated that Fg was less adhered in the HS/DAC-coated catheter than the bare catheter (Fig. 3d). On the contrary, negligible differences were observed in the amount of bovine serum albumin (BSA) by bicinchoninic acid (BCA) assay when samples soaked in PBS solution of BSA (Fig. 3c). It is speculated that this different phenomenon is related to the activation of Fg. Conformational changes in Fg structure during Fg activation to form a fibrin network result in exposure of multiple binding sites. These binding sites can bind to various proteins, causing protein adsorption. Hence, the effect of HS/DAC coating on Fg activation after 2 h of circulation in whole blood was investigated by assessing the exposure of the γ chain via the immunofluorescence staining method. As shown in Fig. 3d, HS/DAC-coated catheters expose significantly less γ chains compared to bare catheters.

Adherent Fg or platelets can promote the adhesion of leukocytes. Leukocytes, in turn, enhance the activation of local platelets and

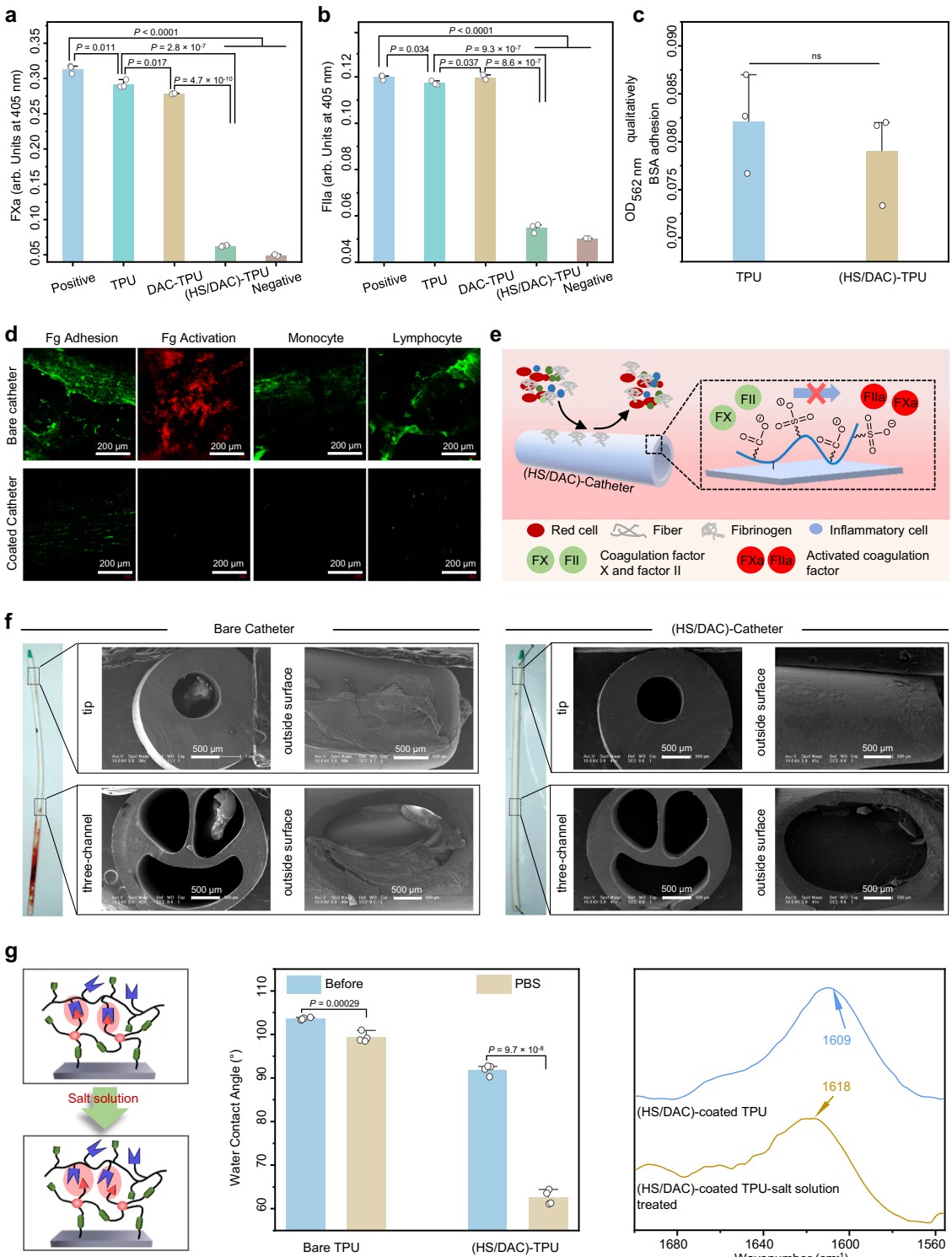

**Fig. 3 | Evaluation of the anti-thrombotic ability of the HS/DAC coating in vitro and in vivo. a** Anti-FXa and **b** anti-FIIa assays of the anticoagulant activity of heparin on the HS/DAC-coated TPU (mean ± SD, $n = 3$ independent samples). **c** Amounts of BSA adsorbed on bare and HS/DAC-coated TPU. The sample was immersed in BSA solution and the adsorption of proteins was determined by a BCA protein assay kit (mean ± SD, $n = 3$ independent samples). **d** CLSM images of Fg adhesion, activation, and leukocytes adhesion on the surface of bare catheters and HS/DAC-coated catheters. Measurements were repeated three times independently with similar results. FITC-Fg was added to fresh whole blood and cycled with samples through a closed circular tube loop for 2 h in the assay of Fg adhesion and activation.

Leukocyte adhesion was detected via immunofluorescent. **e** Schematics of the antithrombotic mechanism of HS/DAC-coated catheters. **f** Digital and SEM images of the thrombus on the bare and HS/DAC-coated CVC catheters in the acute canine model in vivo ($n = 3$ independent samples). Measurements were repeated three times independently with similar results. **g** Process by which the electrostatic interaction of the HS/DAC coating dissociates and the results of WCA (mean ± SD, $n = 4$ independent samples) and ATR-FTIR spectra of HS/DAC-coated TPU before and after treatment with PBS. One-way ANOVA with Tukey's post hoc test was used in (**a**, **b**). Two-sample Student's t test was used in **c**. ns: no significant difference. $P = 0.5$ in c.

participate in the formation of thrombus[41,42]. Therefore, monocytes, lymphocytes and neutrophils, as typical inflammatory cells, were selected to explore the leukocytes adhesion. Benefiting to the strong ability of HS/DAC coatings to reduce Fg activation, a large number of inflammatory cells passively trapped in fibrin clots were observed only on the surface of the bare catheter (Fig. 3d and Supplementary Fig. 8b). In summary, HS/DAC coating can avoid Fg activation due to the inhibition of thrombin and other coagulation cascade proteases, and also reduce the adsorption of inflammatory cells (Fig. 3e). The anticoagulant mechanism indicated that the HS/DAC coating has great potential to reduce thrombosis.

Anticoagulant coatings have important significance and application in various blood-contact medical devices. Firstly, the anticoagulant function of HS/DAC coating was explored by clotting time in vitro. HS/DAC-coated glass bottles have the longest clotting time in recalcified blood compared to bare glass bottles, indicating the potential of the coating for anticoagulant vascular applications (Supplementary Fig. 9a, b). Compared to commercial infection-resistant CVC catheter (Arrowg + ard Blue), the thrombus mass of HS/DAC-coated CVC decreased by 62.5% through a closed circular tube loop of whole blood (Supplementary Fig. 9c). This indicated that the HS/DAC coating has potential advantages in inhibiting thrombosis and avoiding side effects caused by heparin injection methods in the clinic.

To further demonstrate the delayed clotting of HS/DAC-coated catheters, an ex vivo circuit model was carried out. The bare catheters and HS/DAC-coated catheters were implanted in three rabbits via an arteriovenous shunt model (Supplementary Fig. 10a)[43]. In the image of the HS/DAC-coated catheter, a noticeable reduction was observed in thrombi and occlusion formation compared with that of the bare catheter (Supplementary Fig. 10b), and the mass of the thrombus was reduced by 85% (Supplementary Fig. 10c).

Then beagle canines, which are more consistent with acute thrombosis, were selected as animal models to illustrate the antithrombotic properties of HS/DAC-coated commercial CVCs in vivo. Commercial CVC and HS/DAC-coated CVC were aseptically inserted into the left and right jugular veins of three canines, respectively (Supplementary Fig. 11a). After 24 h of implantation, the jugular vein was removed and opened lengthwise. Few thrombi could be observed on HS/DAC-coated CVCs, but substantial visual thrombi were observed on the tip, side hole, and lumen of bare CVCs, which could cause CVCs to fail (Fig. 3f). Compared with commercial bare CVC, HS/DAC-coated CVC reduced the mass of thrombus by 84.6% (Supplementary Fig. 11b).

The long-lasting antithrombotic stability of HS/DAC coating is another crucial for the implantation of medical catheters. Therefore, the antithrombotic stability of HS/DAC-coated TPU and its HS/QAC control was initially investigated via chromogenic assays of anti-FXa and anti-FIIa after incubated in 0.9% NaCl solution of 30 days. Contributing to the strong covalent bonding of HS/DAC coating, the inhibitory activity against FXa and FIIa was maintained well even after one month (Supplementary Fig. 12). The hydrodynamic behavior in the human body is also a challenge to the stability of the coating. Therefore, we further investigated the long-term antithrombotic stability of HS/DAC coatings in artificial blood with shear rates of 30 s$^{-1}$ and 300 s$^{-1}$ (Supplementary Fig. 13a). The HS/DAC-coated catheter can still reduce thrombosis by 61.7% when the shear rate of artificial blood increased to 300 s$^{-1}$ (Supplementary Fig. 13b). Finally, a rat subcutaneous implantation model was established to verify the antithrombotic stability of HS/DAC coating in vivo. Compared with the blank group, the weight of thrombus on the HS/DAC-coated surface was decreased by more than 65% after 30 days of implantation (Supplementary Fig. 13c, d).

The antithrombotic function of the HS/DAC coating may be originated from its self-adaptivity in salt solution. After the gradual dissociation of HS/DAC fixed on the substrate, more functional groups including sulfamate, sulfonate, and carboxyl groups of HS were exposed to realize the antithrombotic performance. To verify this

salt-triggered effect, we examined the physicochemical properties of the HS/DAC coating before and after treatment in salt solution. The self-adaptivity of (HS/DAC)-TPU was characterized by static water contact angles, FTIR and XPS. After incubation with PBS solution, the contact angle of (HS/DAC)-TPU decreased significantly, and the band of the carboxyl group shifted from 1609 to 1618 cm$^{-1}$, which was attributed to the destruction of electrostatic interactions between the carboxyl groups and the quaternary amine groups (Fig. 3g and Supplementary Fig. 14).

## Antibacterial and biocompatibility properties of the HS/DAC coating in vitro and in vivo

CRBIs are another serious complication that increases the risk of mortality with implanted IV catheters[44]. *S. aureus* and *E. coli* were selected as model microorganisms to evaluate the broad-spectrum antibacterial property of the HS/DAC coating. As shown in Fig. 4a and Supplementary Fig. 15, the agar plate colony counting assay revealed that the HS/DAC-coated TPU surfaces exhibit high bactericidal rates (above 99.99% and 99.12%) against *S. aureus* and *E. coli* via a contact-killing mode. SEM and CLSM images were further used to study the morphological changes in *S. aureus* and *E. coli* on (HS/DAC)-TPU. For bare TPU, numerous *S. aureus* and *E. coli* with intact cell morphology are present on the film surface, resulting in a great tendency to form biofilms (Fig. 4b and Supplementary Fig. 16a). In contrast, the attached bacteria on the (HS/DAC)-TPU exhibited shrunken, empty and engulfed morphologies, and some intracellular matter flowed out of the bacteria. This broad-spectrum antibacterial property of the HS/DAC coating can be ascribed to the long lipophilic alkyl chain of DAC, which could penetrate the cell walls and membranes of bacteria, causing autolysis[45]. Similar to commonly used commercial anti-infective CVC catheters, the bactericidal rate of (HS/DAC)-CVC against *S. aureus* was also greater than 99% (Supplementary Fig. 16b).

The antibiofilm activity of (HS/DAC)-TPU was further investigated by extending the coincubation time of the sample with bacteria. As shown in Supplementary Fig. 17a, b, (HS/DAC)-TPU was mostly free from bacterial colonization, and no trace of mature biofilm was found. Compared to the bare TPU, (HS/DAC)-TPU reduced the biofilm coverage by 94.7% at 48 h.

Additionally, bacterial colonization on the lumen surfaces of catheters is another factor that causes CRBIs[18]. Thus, the antibacterial property of the (HS/DAC)-catheter under static and dynamic flow conditions was further demonstrated. In contrast to the bare catheter under static conditions, approximately 99% of the bacteria in the lumen surface of the (HS/DAC)-catheter died (Fig. 4c). For the dynamic flow test, the PVC catheter was simulated as blood vessels, and LB continuously flowed through the catheter that was pre-inoculated with bacteria (Fig. 4d). A great number of bacteria adhered to the lumen surfaces of bare catheters at 24 h, and white biofilms were deposited on the lumen surfaces of the catheter when the flow time was extended to 48 h. However, only a few live bacteria were observed on the (HS/DAC)-catheter and PVC tubes of the (HS/DAC)-catheter channel with a higher transparency (Fig. 4e and Supplementary Fig. 17c).

To simulate in vivo environments, a mouse subcutaneous implantation model of bacterial infection was designed. An indwelling catheter (24 G, TPU) inoculated with *S. aureus* was implanted into two sides of the mouse back (n = 9). After 5 days, a serious purulent phenomenon was observed in the incisions surrounding the bare catheters, while no obvious infection occurred surrounding the HS/DAC-coated catheters (Fig. 4f). Great amounts of live bacteria dispersed in tissue for bare catheter implantation, but few were detected in HS/DAC-coated catheters (Supplementary Fig. 18a). Hematoxylin and eosin (H&E) staining of the tissue surrounding the catheters further demonstrated that extensive neutrophil granulocytes and lymphocytes were present in the bare catheter groups. Proinflammation-associated factors of TNF-α and IL-6 in the tissue were studied to

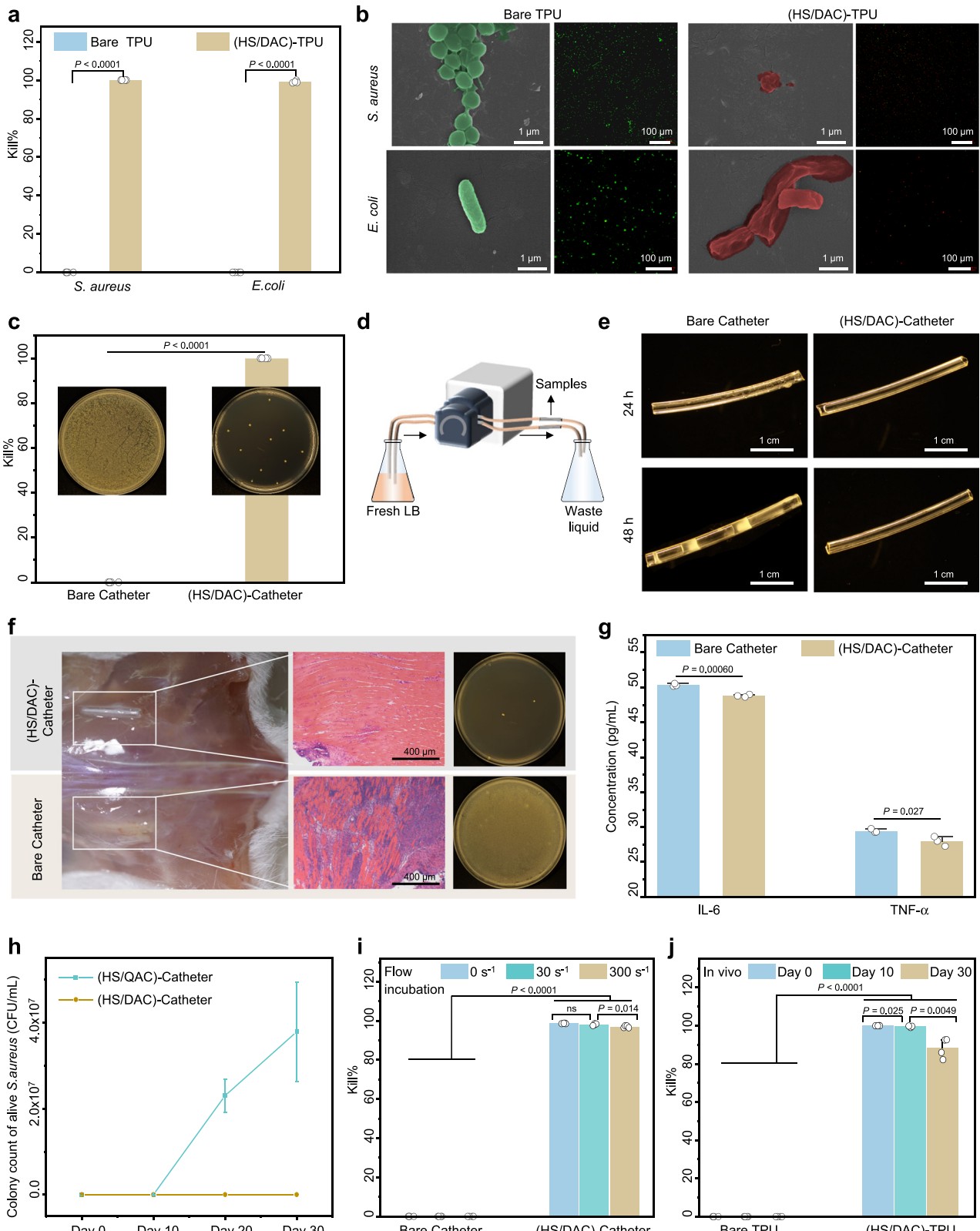

further confirm the inflammation regulation prompted by bacterial infection. As shown in Fig. 4g, the detected factors were significantly reduced in the HS/DAC-coated catheter, which is consistent with the histological assay results.

CVC catheters are usually implanted for approximately 2–4 weeks. Hence, the long-lasting antibacterial property of the HS/DAC coating should also be considered for the effective prevention of catheter-related infection in vitro and in vivo. Compared to the (HS/QAC)-catheter, the (HS/DAC)-catheter still maintained its original antibacterial activity even after one month of treatment in 0.9% NaCl at 37 °C (Fig. 4h and Supplementary Fig. 18b). Next, the long-term antibacterial stability of HS/DAC coating in artificial blood with a shear rate of 30 s⁻¹ and 300 s⁻¹ was investigated by using the multichannel circulation system. As shown in Fig. 4i, the bactericidal rate was still as

**Fig. 4 | Evaluation of the antibacterial property of the HS/DAC coating in vitro and in vivo. a** Antibacterial activity of the HS/DAC coatings against *S. aureus* and *E. coli* via agar plate colony counting (mean ± SD, *n* = 4 independent samples). **b** SEM and CLSM images of bacteria on the surface of bare TPU and HS/DAC-coated TPU films after culture with *S. aureus* or *E. coli* for 24 h at 37 °C. Measurements were repeated three times independently with similar results. **c** Agar plate colony counting assay of the lumen surfaces of catheters under static conditions (mean ± SD, *n* = 3 independent samples). **d** Schematic diagram of the antibacterial dynamic flow test and **e** corresponding images of catheters at different times. **f** Antibacterial property of the HS/DAC coating in vivo, including digital images of infection around the implant site, H&E staining and colony count (mean ± SD, *n* = 3

independent samples). Measurements were repeated three times independently with similar results. **g** Quantitation of cytokines in the tissue at the implantation position (mean ± SD, *n* = 3 independent samples). **h** The amounts of live bacteria on the inner surface of the catheter with the HS/QAC control coating or HS/DAC coating shocked with 0.9% NaCl for 0, 10, 20 and 30 days (mean ± SD, *n* = 3 independent samples). **i** The long-term antibacterial stability of HS/DAC coating in artificial blood with shear rates of 0, 30, and 300 s$^{-1}$ after 30 days of flow (mean ± SD, *n* = 4 independent samples). **j** The antibacterial stability of the HS/DAC coating after implantation in vivo for 0, 10 and 30 days (mean ± SD, *n* = 4 independent samples). Two-sample Student's t test was used in (**a**, **c**, **g**). One-way ANOVA with Tukey's post hoc test was used in (**i**, **j**). ns: no significant difference. *P* = 0.053 in **i**.

high as 97% even if flowing in artificial blood with high shear rate for 30 days. Furthermore, the long-term antibacterial stability in vivo was further explored in a rat subcutaneous implantation model. Agar plate colony counting assay demonstrated that the HS/DAC coating exhibited a good bactericidal rate (>88%) after 30 days of implantation (Fig. 4j).

Biocompatibility is a fundamental requirement for biomedical materials, and we firstly performed a Cell Counting Kit-8 (CCK-8) test to determine the cytotoxicity of HS/DAC. As shown in Supplementary Fig. 19a, cell viability was greater than 90% regardless of whether the extracts of (HS/DAC)-TPU or (HS/DAC)-TPU contact with the cells. However, (HS/QAC)-TPU control showed obvious cytotoxicity, especially as low as 5% when in direct contact with cells. These above results indicated that the formation of crosslinked networks in HS/DAC coating can greatly improve cell compatibility. To further illustrate the biocompatibility in vivo, the histocompatibility of bare and HS/DAC-coated catheters was investigated by a subcutaneous implantation model in six mice and a vasculature model in three acute canines. The HS/DAC-coated catheter maintained similar inflammatory levels to the bare catheter via H&E staining, indicating no obvious inflammation (Supplementary Fig. 19b). Negligible changes in IL-6 and TNF-α further confirmed the good biocompatibility of the HS/DAC-coated catheters in mice (Supplementary Fig. 19c). Additionally, histopathological analysis of the vasculature of canine models via H&E staining indicated that implanted catheters in blood vessel did not cause an obvious inflammatory reaction (Supplementary Fig. 19d).

## HS/DAC coating against catheter-related bacteremia

The reciprocal feedback pattern established between the biofilm and the thrombus can lead to serious diseases[14]. Typically, bacteria introduced during catheter implantation can influence individual coagulation and activate blood clotting by releasing a variety of exotoxins to induce the activation and aggregation of platelets and smooth muscle contraction (Supplementary Fig. 20a). However, few catheter-related bacteremia and catheter infection-related thrombi are considered. Therefore, an ex vivo circulation model of closed-loop catheters in rabbits was established to demonstrate the better performance of our HS/DAC coating (Fig. 5a). The amount of thrombus on the bare catheter was much higher than that on the HS/DAC-coated catheter in four rabbits, especially on the bare catheter inoculated with bacteria (Fig. 5b). The area of thrombus was counted by ImageJ V1.49 software. As shown in Fig. 5c, the HS/DAC-coated catheter reduced the thrombosis-blocking rate by 92.7%, indicating that the HS/DAC coating could effectively prevent thrombus formation regardless of the presence of bacteria. Digital images and SEM images of the cross-section also showed a massive macroscopic thrombus on the bare catheters. In particular, the occlusion of the lumen was serious in the bare catheter with bacteria (Fig. 5d i and ii). These results suggested that the presence of bacteria on the catheter may accelerate the formation of thrombi. Subsequently, the adhesion of bacteria on the surface of catheters and in the circulation was investigated by colony counting. A large number of bacteria can be clearly observed on the bare catheter inoculated with bacteria. Meanwhile, a severe thrombosis and bacteria

were detected in the blood of the circulation line of bare catheter inoculated bacteria (Fig. 5d iii and Supplementary Fig. 20b, d). In addition, compared to the thrombus formed on the bare catheter, the bare catheter inoculated with bacteria promoted the formation of fiber and fibrin sheaths (Fig. 5d iv), which are associated with biofilm formation[18,46,47]. The coagulation system is activated and forms immunothrombosis by the interaction of innate immune cells and platelets to limit the spread of bacteria in the bloodstream when pathogens invade the bloodstream[14]. The attachment of monocytes on catheters was further tested by immunofluorescence detection. Many monocytes were found in bare catheters, while few cells were found in HS/DAC-coated catheters. Additionally, there were more monocytes in the bare catheter with bacteria (Fig. 5d v). Overall, these very promising results demonstrated that the HS/DAC coating has the potential to effectively prevent the formation of immunothrombosis and catheter-related bacteremia.

## Discussion

Life-threatening thrombosis and bacterial infections caused by cardiovascular catheters are significant clinical challenges[1,48]. Several antithrombotic or antibacterial strategies, classifying into noncovalent coating and covalent grafting, have been used to modify cardiovascular catheters[49,50]. However, their applications are limited to time-consuming preparations[12,51–53] and poor long-term stability under complex physiological environments[54,55].

In contrast to those methods mentioned above, our HS/DAC coatings may provide an available resolution for this issue. As-synthesized liposoluble HS/DAC complex realizes the one-step coating on various medical devices with different compositions, sizes, and shapes without changing the transmittance of substrates, which shows great potential for large-scale practical usages. Meanwhile, the formed cross-linked network within HS/DAC coating significantly enhances the structural stability as demonstrated under various extreme conditions in vitro and also long-term performance in vivo.

Furthermore, commercial functional catheters usually have a single bioactivity, such as medical catheters loaded with antibiotics for antibacteria or modified with heparin sodium for antithrombotic usage. Therefore, it is difficult to solve bacterial infections and thrombosis simultaneously. According to Debye-Hückel theory, the electrolyte in physiological environments could weaken the electrostatic dipole-dipole interactions between HS and DAC, leading to the adaptive dissociation feature to meet the requirements of both antithrombotic and antibacterial capabilities. The strong antithrombotic function of this adaptable HS/DAC coating is attributed to the cutting off coagulation cascade, inhibiting Fg activation and inflammatory cell participation. In this way, the HS/DAC coating can reduce the thrombus by at least 60% on the outer and inner surfaces of IV catheters in vitro and in vivo. Moreover, in vitro and in vivo studies demonstrated that the HS/DAC coating can effectively kill more than 97% of bacteria by a contact-killing mode. Compared with commercially available anti-infective CVC catheters (Arrowg + ard Blue), HS/DAC-coated CVC catheters have comparable antibacterial properties and the amount of blood clot is reduced by 62.5%. More importantly,

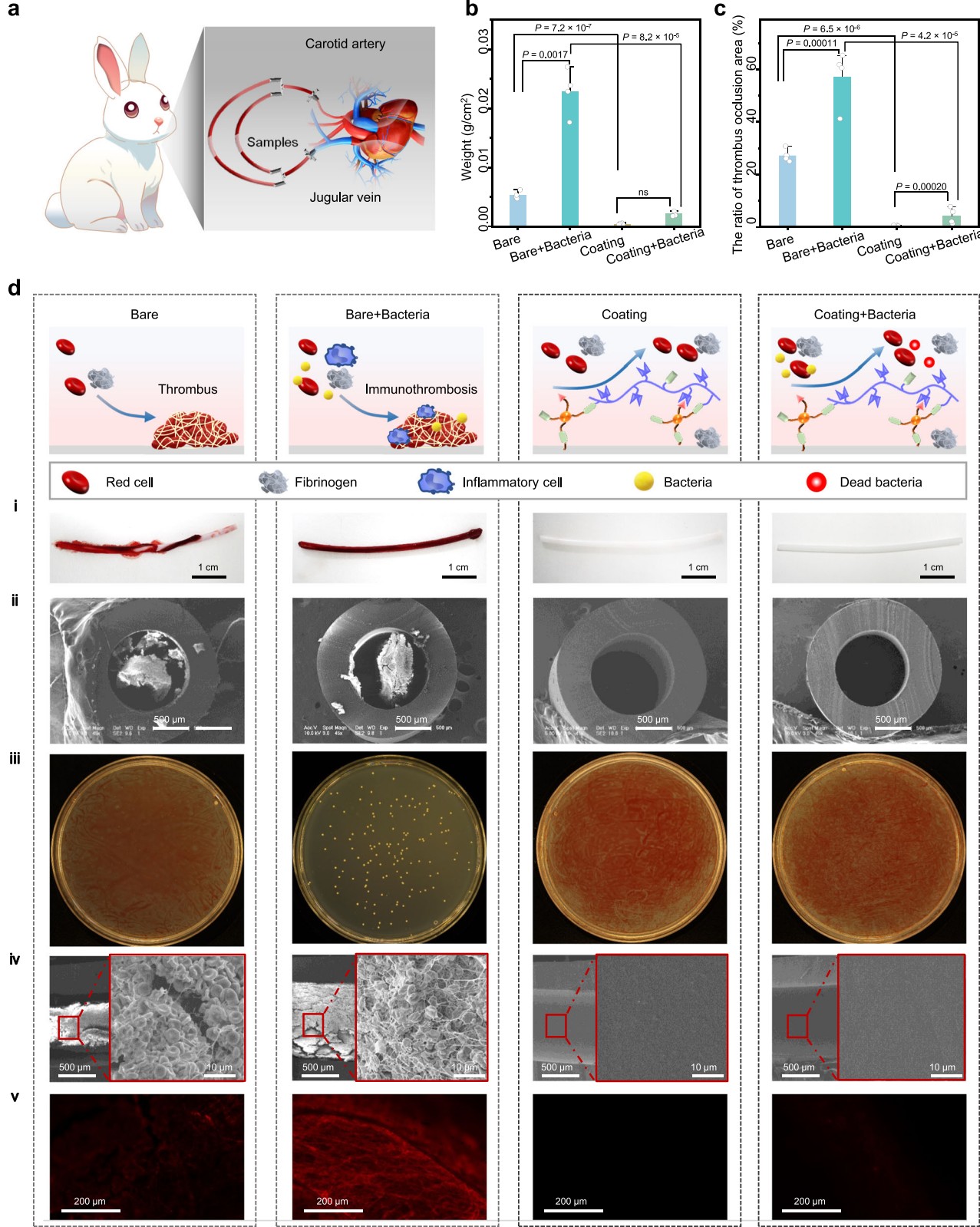

**Fig. 5 | The good performance of the HS/DAC coating in inhibiting catheter-related bacteremia and bacterium-related thrombi. a** The ex vivo rabbit model of catheter-related bacteremia and bacterium-related thrombi. **b** Changes in catheter weight after circulation (mean ± SD, $n = 4$ independent samples). **c** The extent of occlusion in the bare catheter, infected bare catheter, HS/DAC-coated catheter and infected HS/DAC-coated catheter (mean ± SD, $n = 4$ independent samples). **d** Schematics showing the bacterium-related mechanism of thrombus formation on infected catheters in vessels and the results of the ex vivo test. Digital images (**i**), SEM images of cross-sections (**ii**), optical images of live bacteria in blood after circulation (**iii**), SEM images of inner surfaces (**iv**) and immunofluorescence detection of monocytes (**v**) of the bare catheter, infected bare catheter, HS/DAC-coated catheter and infected HS/DAC-coated catheter. Measurements in (**ii, iv, v**) were repeated three times independently with similar results. One-way ANOVA with Tukey's post hoc test was used in (**b, c**). ns: no significant difference. $P = 0.052$ in **b**.

the formation of infection-related thrombi and catheter-related infection was significantly reduced in an ex vivo rabbit model.

Consequently, in virtue of these satisfactory capabilities, this complex has the potential to supply a facile and efficient strategy to build functional coatings for long-term use against catheter-related infection and thrombi. In addition to cardiovascular catheters, HS/DAC may become a coating material for other vascular implants, including artificial valves and stents. More importantly, HS/DAC can be modularly designed and modified for more functional coatings. In brief, we expect that this coating strategy will find wide applications in biomedical materials.

## Methods

### Preparation of HS/DAC-coated samples

Medical-grade samples, including flat surfaces, tubing or other shapes, were cut to the desired size, cleaned with ethanol and deionized water three times, and fully dried. All the samples were air-plasma treated for 5 min to obtain hydroxyl groups. Glass was hydroxylated with piranha solution following our previously described procedure[56]. Briefly, glass was immersed in a piranha solution (consisting of concentrated sulfuric acid and 30% hydrogen peroxide in a volume ratio of 3:1) and boiled for 1 h. Subsequently, samples were submerged in HS/DAC dichloromethane solution (10 mg/mL) for 1 min to construct HS/DAC-coated samples. The HS/DAC-coated samples were dried at room temperature to evaporate dichloromethane. After that, the samples were washed 3 times with deionized water and cured at 100 °C for 30 min. Finally, the HS/DAC-coated samples were ultrasonically cleaned 3 times in deionized water and dried at room temperature.

### Stability of the HS/DAC coating

**Chemical durability of the coating.** The control samples of HS/QAC complex and HS/QAC-coated TPU were obtained by the same methods as above. To determine the stability of the coating in solution, (HS/QAC)-TPU and (HS/DAC)-TPU films were immersed in a variety of solvents, including ethyl acetate, n-hexane, methyl glycol, glycerol (30 v/v % in water) and 75 v/v% alcohol, salt solution (physiological saline, 30 wt % NaCl solution and PBS), and NaOH/HCl solution with different pH values (from 4 to 9), followed by strong ultrasonication (300 W, 40 kHz) for 3 h. After that, TPU before and after treatment was stained with Rose Bengal and fluorescein sodium salt for further visualization.

**Mechanical stability of the coating.** HS/QAC and HS/DAC-coated catheters were simultaneously manually bent to 180° for 500 times. The morphological changes of the coatings after bending were characterized with SEM.

**Stability of the coating under extreme conditions.** Samples were immersed in liquid nitrogen for 1 h or treated with high temperature and high pressure (120 °C, 103.4 kPa) for 30 min. Then, the morphological changes of the coatings were monitored with SEM.

**Stability of the coating in salt solution under flow incubation.** TPU films (4 cm × 0.3 cm) with or without coating were placed in the independent silicone tube channel with an inner diameter of 0.3 cm. Subsequently, a peristaltic pump (LEAD FLUID, BT101 L) was utilized to connect the silicone tube. Normal saline was injected into the peristaltic pump to form a closed loop system, and a flow rate of 38.4 mL/min was set to simulate venous blood flow. Bare films and HS/DAC-coated films (0.5 cm × 0.3 cm) at 0, 10, and 30 days with flow treatment were soaked in Rose Bengal solution (9.8 mM) for 3 min, removed and rinsed with ultrapure water. The peeling of the coating was recorded through digital photographs. In addition, the dye adsorbed on the coating was redissolved in ethanol, and its absorption strength at 548 nm was measured.

**Stability of the coating in vivo.** All animal procedures were performed in accordance with the Guidelines for the Care and Use of Laboratory Animals of the Chinese Academy of Sciences and approved by the Animal Ethics Committee of Changchun Institute of Applied Chemistry, Chinese Academy of Sciences (no. 20210027). Female Sprague-Dawley rats (SD rats, 160–200 g, 7–8 weeks old) were housed at an ambient temperature of 25 °C (24–26 °C), humidity of 30% and a photoperiod of 12 h light/12 h dark. In addition, a standard diet and water are given. HS/DAC-coated or bare TPU films with a diameter of 0.8 cm were implanted into the subcutaneous tissue on both sides of the back of SD rats. After 10 and 30 days of implantation, samples were removed and washed. The staining experiment was conducted as described above.

### Anti-infection-related thrombus and catheter-related bacteremia of HS/DAC coating

The ex vivo circulation was built as an arteriovenous shunt model following the procedure reported[57,58]. Four adult New Zealand white rabbits (3.0–3.5 kg) were general anaesthetized. The left carotid artery and the right external jugular vein of rabbits were isolated and connected to build the ex vivo circulation. All the materials were sterile. Bare catheters and HS/DAC-coated catheters that were sterilized without bacterial treatments were named Bare and Coating, respectively. Bare catheters and HS/DAC-coated catheters that were incubated with *S. aureus* (10⁶ CFU/mL) for 3 h at 37 °C were named Bare + Bacteria and Coating + Bacteria. All four samples were tested on the same rabbits to minimize the systemic errors. In order to avoid the influence of bacteria, we first assembled Bare or Coating samples into the rabbit's arteriovenous shunt. Once the catheter becomes blocked, the experiment is terminated and the catheter access is carefully removed. At the same time, the assembled circulation containing the Coating + Bacteria and Bare + Bacteria samples were connected to the arteriovenous shunt separately. The samples were gently washed 3 times with PBS and weighed. Thrombus formation on the lumen and surface was observed by digital camera and SEM. In addition, catheters and blood were collected from each circulation line to detect viable bacteria adhering on the surface and in the blood. Briefly, catheters were immersed in PBS solution for ultrasonic treatment. After 5 min, 200 μL of solution were taken for colony counting. A section of tubing (~5 cm) adjacent to the catheter samples was intercepted with a hemostatic clip. Blood or clots were then removed from the tube (~1 cm) and 1 ml of PBS is added for ultrasound and colony count. Immunofluorescence detection of monocytes on the surface of different sample groups was observed by CLSM. Catheter segments were fixed in 4% of paraformaldehyde and rinsed with PBS. These segments were incubated in primary antibodies against monocytes (anti-CD14, 1:1000) for 1 h. Samples were rinsed with PBS and further incubated for 1.5 h in fluorescently conjugated secondary antibody (lgG (H + L), Alexa Fluor 555-labeled Donkey Anti-Rabbit, 1:200). Finally, stained tubes were rinsed and freeze-dried and observed by CLSM.

### Statistical analysis

Each experiment was performed with at least three replicate measurements. Data are presented as the means ± standard deviations (means ± SD). Statistical analysis was performed using Origin 2020 software and Microsoft Excel. One-way analyses of variance (ANOVA) followed by Tukey's post hoc test were used for the statistical analysis between multiple groups, and two-sample Student's t-test was used for comparisons between two samples.

### Reporting summary

Further information on research design is available in the Nature Portfolio Reporting Summary linked to this article.

## Data availability
All data are available in the main text, Supplementary Information, or Source Data file. Source data are provided with this paper. If any raw data files are needed in another format, they are available from the corresponding author upon request. Source data are provided with this paper.

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

## Acknowledgements

This work was supported by the National Natural Science Foundation of China (Grant No. 51973221, H.S., 52293384, S.L.), Youth Innovation Promotion Association of CAS (Grant No. Y2021066, H.S.), National Key Research and Development Program of China (Grant No. 2021YFC2101700, X.Z.), Scientific and Technological Development Program of Jilin Province (Grant No. 20220508090RC, H.S.), and High-Tech Research & Development Program of CAS-WEGO Group (S.L.).

## Author contributions

S.L. and H.S. proposed and supervised the project. L.L. and H.Y. designed the experiment and participated in the entire project. X.Z. assisted in the establishment of animal experiments. L.L., H.Y., L.W., and H.S. wrote the manuscript. L.W., S.L., and H.S. contributed to discussions on the results. L.W., D.Z., X.D. J.Y. and H.S. contributed to the visualization. All authors have given approval to the final version of the manuscript.

## Competing interests

The authors declare no competing interests.
