## [Peer Review File · Nature Communications]

Heparin-Network-Mediated Long-lasting Coatings on Intravascular Catheters for Adaptive Antithrombosis and Antibacterial InfectionREVIEWER COMMENTS

Reviewer #1 (Remarks to the Author):

This manuscript introduces a preparation method of antibacterial and anticoagulant coating. Through the electrostatic interaction between quaternary ammonium salt and heparin, a complex was formed, which is then dissolved in organic solvent, and then the coating is prepared by dipping method. Systematic *in vivo* and *in vitro* tests have been carried out, and the antibacterial and anticoagulation ability of the coating is good. However, the coating preparation method has been previously reported, no breakthrough was found in this manuscript, and I do not recommend that this manuscript to be published in Nature Communications.

1. Will siloxane react with the hydroxyl group of heparin? The author shows no reaction in Figure 1. Why not?
2. After heparin is combined with cationic quaternary ammonium salt, why can cations play a bactericidal role? Only hydrophobic chain is not bactericidal. If it is because in ionic solution, cations became "free condition", play a bactericidal role, then why the anti-coagulation and anti-adhesion properties could be accomplished in this condition? The reverse is the same question, while heparin immobilized, the anionic charge locked up so much, why the anti-coagulation still accomplished? Although the authors achieved simultaneous antibacterial and anticoagulant properties, the mechanism has not been fully clarified by the authors. Reviewer believe that one of the key points may be the ratio of positive and negative charges, as well as the ratio of free ionic groups to non-free ionic groups that bond under physiological conditions. This needs to use spectroscopy experiment.
3. The reason why the anticoagulant and bactericidal properties can be maintained at the same time in this paper is not clarified, and the anticoagulant and antibacterial properties are not discussed combined with the characteristics of the material. In the discussion part of anticoagulant and antibacterial tests, the material and biological properties discussions are separated, and the analysis of biological properties should end with the discussions of the material.
4. The method of materials reported in this paper is similar to the previous reports of the author, which is to prepare cation and anion complex and then make coating by dipping coating. This paper does not show breakthrough progress, but only introduces silane coupling agent. The experiments in this paper are very sufficient, but no breakthrough is found in the material.
5. The author first made bacteria adhere to the surface of the catheter, and then conducted blood tests to prove that bacterial adhesion can promote coagulation. It is suggested that the authors add the opposite test, exposing the material to blood and then conducting antibacterial tests, to show that the clots promote bacterial colonization or reduce the bactericidal properties of the material. Because the author mentioned in the preface that the antibacterial and

anticoagulant are mutually reinforcing, the experiment in one direction is not enough to verify the author's view.

6. In Figure D5, there is a spelling error, bacteria is written as Bactereria, and it appears twice.

Reviewer #2 (Remarks to the Author):

In this work, authors present a coating for catheters, its synthesis and the advantages it presents compared to uncoated catheters, which include antimicrobial and antithrombotic activity, and study biocompatibility of these modified catheters. Moreover, authors test the coatings resistance and properties under various stress conditions, obtaining excellent results.

The topic seems novel, and the new material appears to be an important solution to various health problems regarding complications due to catheter use in patients. Work is extensive and sound and most conclusions are well supported by it.

Paper could benefit from a proof reading by a native English speaker, as redaction is a bit unclear at times, especially in the introduction and results portions, and some colloquial language is used.

Moreover, distribution of information throughout the paper is confusing: the results carry most of the discussion, whereas the discussion seems more like conclusions or a summary of the results. Discussion would be richer if some comparison with other types of coatings was presented. Some methods have text that I would categorize as introduction or discussion.

Below, some comments and clarifications that could benefit the paper:

Introduction

It is stated that "90% of patients require IV catheters". Does this refer to cancer patients? Would need a reference.

I'm unclear as to what "20-400 thousand of mortality" means. Is it number of people? In the world? In which period of time?

Results

The paper states "(HS/DAC)-TPU exhibited obvious inhibition of FXa and FIIa" and references a supplementary figure. It would be interesting if these results were further commented, either qualitatively or quantitatively.

BCA assay and leukocyte adhesion have no methods or references.

The colony count assay as explained (sonication of bacteria in catheter, suspension and distribution in an agar plate) gives a notion of how many bacteria remained adhered to the surface, not necessarily of the coating's ability to inhibit or kill them. In the contact inhibition assay, both catheters (coated and uncoated) seem to have the same inhibition capacity, restricted only to bacteria directly in contact with the material. These two assays combined don't seem enough to conclude that the coating has an antimicrobial ability in a contact-killing mode.

Antibiofilm activity has no method per-se, it's briefly redacted in results.

SEM shows cell adhesion and morphology. When looking at *E. coli* in bare TPU, a biofilm seems to have formed, but for *S. aureus*, cells have adhered, but the presence of a biofilm is not clear in images. Therefore, this experiment may not lead to the conclusion that coated TPU is able to inhibit biofilm formation in comparison to uncoated one, at least for *S. aureus*. Later on, the authors refer to a visible biofilm formation in bare catheters and not in coated ones. Moreover, SEM images showed that coating changes bacterial morphology, which can affect biofilm formation. All of this information combined could lead to a sound theory of the coated material's ability to inhibit biofilm formation, but does not seem conclusive.

Methods

In the stability assay, triglyceride (30 v/v% in water) is mentioned as a solvent. I'm not sure it is, or if it can be dissolved in water.

In Anti infection-related thrombus and catheter-related bacteremia of HS/DAC coating, the use of three rabbits is mentioned, but then four different groups appear. Seems like each group had three rabbits, but redaction is unclear.

In antibacterial properties in vivo (supplementary), it's commented that "catheters were sterilized by immersing in 75% ethanol for 30 min". Ethanol is not a sterilizing agent, but rather a disinfecting one. Antibacterial assays have no positive controls reported.

Clotting time method is explained in supplementary materials but has no results.

Reviewer #3 (Remarks to the Author):

Review to Ms NCOMMS-22-43549A-Z

L. Liu, Heparin-Network-Mediated Long-lasting Coatings on Intravascular Catheters for Adaptive Antithrombosis and Antibacterial Infection

The manuscript describes an antithrombotic and antimicrobial coating made of heparin and a silane with a quarternary ammonium group. The coating binds firmly to various medical devices, such as catheters, after a hydroxyl-group forming activation step of the surface. The authors provide structural analysis of the coating and functional in vitro and animal tests.

The presentation of the study would need intensive restructuring. The authors performed and present multiple experiments, but do not introduce the principal setup and specific question of the experiment in the text. Multiple different analyses are presented in a single paragraph without any logical sectioning. Important information, such as the full name of DAC, a main coating component, is found only in the supporting information. – The language is well understandable but would need careful revision. A more restrictive use of abbreviations would simplify reading.

This type of presentation strongly impedes careful scientific review. The reviewer would be very willing to review a reorganized manuscript again. One conceptional problem can be indicated here:

The antimicrobial action of quarternary ammonium salts and the anticoagulant activity of heparin are both closely linked to their charge. The complex formation in this coating neutralizes these charges. The

authors phenomenologically demonstrate high (residual) activity. However, this should be rated to the activity of the native compounds.

Reviewer #4 (Remarks to the Author):

This important work addresses the need for antithrombotic and antimicrobial catheters, by developing novel heparin-containing polyelectrolyte complexed coatings on polyurethane catheters. There are some important items that should be fixed prior to publishing this work.

1. Three different animal models were used in this work (rabbits, rats, and canines). The authors do not offer justification for the selection of these animal models. They also do not indicate the number of animals used (for the canine studies) or the justification for the number of animals used (e.g. power calculations) determining how the number of animals in each group was selected. Whether blinding and randomization were used for each study are also not reported. I cannot recommend publication of this work without sufficient justification. I recommend that the authors consult and reference standardized/recommended ethical guidelines for designing and reporting these experiments, such as the ARRIVE guidelines. (<https://doi.org/10.1371/journal.pbio.3000411>)
2. The researchers conducted molecular dynamics simulations, but there is no discussion of how the results of these simulations impacted the study design or the interpretation of the results. This should be either made relevant to the reported work or it should be removed from this report.
3. When reporting the catheter weight, following the rabbit circulation model, is this the total weight of the catheter, or the change in the weight associated with the thrombus formation (fig. S12). This should be clarified and interpreted. Also this should somehow be normalized to the contact area of the blood to the inside surface of the catheter. An absolute mass for the thrombus without some relative measure of the area or size over which the thrombus formed is not very meaningful.
4. For the coating stability study using flow, the authors should calculate the shear rate at the surface, and justify the use of (Newtonian?) saline solution versus non-Newtonian blood (under the viscosity and shear rate ranges used). These values should also be compared to physiologically relevant ranges of shear rates encountered during catheter use.
5. The entire manuscript should be edited for English language, grammar, and usage. Some errors obscure the meaning.

Point-to-point response:

Reviewer #1

This manuscript introduces a preparation method of antibacterial and anticoagulant coating. Through the electrostatic interaction between quaternary ammonium salt and heparin, a complex was formed, which is then dissolved in organic solvent, and then the coating is prepared by dipping method. Systematic *in vivo* and *in vitro* tests have been carried out, and the antibacterial and anticoagulation ability of the coating is good. However, the coating preparation method has been previously reported, no breakthrough was found in this manuscript, and I do not recommend that this manuscript to be published in Nature Communications.

Our response: Thank you for your comments. Catheter-related bloodstream infections and catheter-related thrombosis have become a life-threatening complication to human health. In this work, we report a concise approach to construct a long-lasting and adaptable coating that combines high antithrombin nature of HS and excellent antibacterial DAC for treating CRBIs and CRT simultaneously. And we believe that our coating approach shows various advantages in comparison with those reported methods:

(1) Facile coating approach without substrate dependence. Different from the conventional one-time coatings and time-consuming covalent grafting coating, our HS/DAC complex can attach on diversity of catheters with arbitrary shapes and compositions *via* facile dip (Fig. 2e and Supplementary Fig. 5 in revised manuscript and Supplementary Information). Also, some biological properties, for example, antibacterial and antithrombotic properties, can be introduced into obtained coating simultaneously. This may supply a potential possibility for large-scale practical application.

(2) Long-term stability and bioactivity under complex physiological environments. Typically, covalent grafting method always needs rigorous process to obtain robust coating, while noncovalent coating suffered from shedding in physiological conditions. In this work, the formation of cross-linked networks between HS/DAC coatings and substrates could greatly enhance the stability in different solvents even under ultrasonic cleaning (3 h), mechanical bending (500 times), and other extreme conditions (in liquid nitrogen and autoclaving) (Fig. 2f and Supplementary Fig. 6 in revised manuscript and Supplementary Information). After

30 days of treatments *in vitro* and *in vivo*, the HS/DAC coating still maintained about 97% of bactericidal activity, and its thrombus adhesion rate also decreased significantly (Fig. 4i-j and Supplementary Fig. 13 and 18).

(3) Salt triggered adaptive dissociation to treat CRBIs and CRT simultaneously.

Currently, there have no commercially available methods to solve CRBIs and CRT due to the complexity and interrelation in blood-contact medical devices. Contributing to the adaptive dissociation feature, HS/DAC coating not only significantly reduced thrombus adhesion by 84.6% in an acute canine model, but also exhibited antibacterial performances by the implantation of catheters in ex-vivo rabbit model (Fig. 5 and Supplementary Fig. 11), which go beyond existing works and also the commercial infection-resistant CVC catheter (Arrowgard Blue, 16 Ga, Teleflex®).

Therefore, such a substrate-independent and portable long-lasting multifunctional coating meets the growing demands against catheter-related infection and thrombi in clinic.

1. Will siloxane react with the hydroxyl group of heparin? The author shows no reaction in Figure 1. Why not?

Our response: Surely, the siloxane can react with the hydroxyl group of heparin. To make this more clearly, the formed Si-O-C bonds is marked with a dotted circle in Figure R1.

Figure R1 Illustration of the preparation of the HS/DAC complex and HS/DAC coating.

2. After heparin is combined with cationic quaternary ammonium salt, why can cations play a bactericidal role? Only hydrophobic chain is not bactericidal. If it is because in ionic solution, cations became "free condition", play a bactericidal role,

then why the anti-coagulation and anti-adhesion properties could be accomplished in this condition? The reverse is the same question, while heparin immobilized, the anionic charge locked up so much, why the anti-coagulation still accomplished? Although the authors achieved simultaneous antibacterial and anticoagulant properties, the mechanism has not been fully clarified by the authors. Reviewer believe that one of the key points may be the ratio of positive and negative charges, as well as the ratio of free ionic groups to non-free ionic groups that bond under physiological conditions. This needs to use spectroscopy experiment.

Our response: Reviewer is right! The ratio of positive and negative charges and their bonding situations under physiological conditions are of great significance to the simultaneous antibacterial and anticoagulant properties. To illustrate the mechanism clearly, several spectroscopic characterizations and antibacterial and anticoagulant experiments have been carried out. Under the simulated physiological condition, formed HS/DAC coating can take place the salt-triggered adaptive dissociation to expose the sulfonate and quaternary ammonium ions, benefitting to maintain the broad-spectrum antibacterial and antithrombus properties. The detail discussions about this mechanism are given in the following and also the revised manuscript.

(1) For the correlation between charge ratio and bio-performances, we initially tried to determine the exact ratio of positive and negative charges within HS/DAC complex by elemental analysis, but failed due to the uncertain saccharide sequences and molecular weights of this natural HS polysaccharide (Carbohydrate Polymers, 2023, 308, 120649; Journal of Medicinal Chemistry, 2003, 46, 2551). Fortunately, the charge ratio of HS/DAC complex can quantify by using the potential characterization. As shown in Figure R2a-b, the zeta potential of HS/DAC complex is determined to be -40.9 mV, and a much lower negative potential is up to -66.8 mV after coating formation. This indicates the formation of desired HS/DAC complex with excess amount of HS. More importantly, our present results demonstrated that this charge ratio was not affect the antibacterial and antithrombus properties of the HS/DAC complex and its coating both *in vitro* and *in vivo*. To testify it, DA/QAC complex with nearly neutralizing charges was synthesized as model to simulate the fully ions shielded situation. And the diversity of antibacterial properties was studied by detecting the minimum inhibitory concentration (MIC) of QAC, DA, and DA/QAC, respectively. As shown in Figure R2c, the calculated MIC value of DA/QAC was at the same level as that of free QAC. This confirms that the electrostatic bonding does not affect the biological properties of the material itself.

Figure R2 (a) Zeta Potentials of free HS, DAC and HS/DAC complex. (b) Surface potentials of TPU films with and without coatings. (c) Molecular structures and *in vitro* antimicrobial properties of DA/QAC model complex with neutralizing charge.

(2) For salt-triggered adaptive dissociation, this similar mechanism has been confirmed in previous works including the small molecular salt ions and polyelectrolyte complexes (for example, *Macromolecules* 2022, 55, 978; *Phys. Rev. Lett.* 2010, 105, 208301; *Macromolecules* 2020, 53, 102). The breaking of associated ion pairs in the electrolyte medium is governed by the activation energy, E_a :

$$E_a \approx E_{corr} - E_{coul} = \kappa l_B \propto \sqrt{c_{salt}} - \frac{l_B}{d}$$

where E_{corr} is the correlation energy of separated charged groups, E_{coul} is the Coulomb energy of ion pairs in the bound state, κ is the inverse Debye length, l_B is the Bjerrum length, d is the distance between two ionic groups when they are bound, and c_{salt} is the solution salinity. It is seen that the activation energy barrier decreases with the increase of salt concentration. In this work, salt ions could cause electrostatic dissociation of HS/DAC ion pairs, thereby exposing carboxylate and sulfonate fragments from HS and DAC. Catheter surfaces with more sulfonic acid and carboxylate ions may benefit to form hydration layers *via* bonding with water molecules, reducing the adhesion of bacterial and blood cells (*Biomaterials*, 2020,

262, 120336; *Acta Biomaterialia*, 2019, 87, 108; *Carbohydrate Polymers*, 2018, 184, 408).

Water contact angle (WCA), FTIR and XPS characterization were conducted to detect the salt-triggered dissociation feature. The coatings were soaked in the physiological environment (0.01M PBS, pH 7.4 or 0.9% NaCl) for 20 min and then dried under nitrogen. As shown in Figure R3a, the WCA of the (HS/DAC)-TPU decreased significantly after incubation with PBS solution. The band of carboxyl group shifted from 1609 to 1618 cm^{-1} , and the sulfonic acid group decreased and shifted significantly, attributing to the destruction of electrostatic interactions between the anion group and the quaternary amine group (Figure R3b). XPS spectra show that the S 2p peaks are shifted from 167.9 and 168.95 eV for the HS/DAC coating to 168.15 and 169.40 eV for the salt treated HS/DAC coating (Figure R3c), respectively. Additionally, the N 1S peak is slightly shifted from 402.05 to 402.15 eV after salt treatment, demonstrating the dissociation between HS and DAC. Meanwhile, benefiting from dissociation of the ion pairs in the HS/DAC coating under physiological environment, the exposed ion groups and enhanced hydrophilicity were more conducive to the inhibition of bacterial infection and thrombogenesis simultaneously. This in turn confirmed the salt triggered adaptability of the coating (Figure R3d).

Figure R3 The water contact angles (a), ATR-FTIR spectra (b) and XPS (c) of bare films and HS/DAC coated films before and after treated with saline solution ($n = 6$). (d) Salt-triggered adaptive dissociation and its antibacterial and antithrombotic mechanism of HS/DAC coating. $***p < 0.001$.

3. The reason why the anticoagulant and bactericidal properties can be maintained at the same time in this paper is not clarified, and the anticoagulant and antibacterial properties are not discussed combined with the characteristics of the material. In the

discussion part of anticoagulant and antibacterial tests, the material and biological properties discussions are separated, and the analysis of biological properties should end with the discussions of the material.

Our response: The discussion of antibacterial and anticoagulant mechanisms is described in the response of above Q2. In addition, we reorganized discussions sections of the material and biological properties in revised manuscript.

4. The method of materials reported in this paper is similar to the previous reports of the author, which is to prepare cation and anion complex and then make coating by dipping coating. This paper does not show breakthrough progress, but only introduces silane coupling agent. The experiments in this paper are very sufficient, but no breakthrough is found in the material.

Our response: Thank you for your interest in our works. Our previous works (for example, *Chem. Eng. J.* 2019, 360, 1030; *ACS Appl. Mater. Interfaces.* 2018, 10, 45, 39257) mainly focus on developing a similar approach to obtain one-step dip coating, but limiting in the weak long-term stability and antibacterial property alone. Hence, we believe that the present work shows great improvements as listed in following:

First of all, Long-term stability. Implantation time and complex working environments of medical catheters, such as high-speed flowing blood and salt-rich urine, make much higher demands on the long-term stability of the coating. The long-term stability of the coating in our previous works is not good enough. These coatings formed with non-covalent bond are more likely to fall off in salt solution, especially the solution with high ionic strength, resulting in coating shedding and catheter failure. In addition, these coatings contained lipid soluble complexes can still be re-dissolved in organic reagents in the catheter applications and disinfection processes, which challenge the effectiveness of these coatings. In this work, the formation of cross-linked networks between HS/DAC coatings and substrates after heating could great enhance the stability even under ultrasonic cleaning (3 h) in different solvents, mechanical bending (500 times), and other extreme conditions (in liquid nitrogen and autoclaving) (Fig. 2f and Supplementary Fig. 6 in revised manuscript and Supplementary Information).

Second, Long-term biological performances. Our work only introduces the silicone quaternary ammonium surfactant to achieve the improvement of the stability performance of the coating, without the additional need for other complicated and tedious processes. Under the simulated body fluid environment, the antibacterial

property of HS/QAC for comparison decreased significantly after 20 days of immersion, while the antibacterial rate of HS/DAC coating was still close to 100% (Fig. 4h in revised manuscript).

Last but not least, our previous works only focused on antibacterial activity, which is hard to satisfy the catheter-related bloodstream infections and thrombosis of blood contacting medical devices. Contributing to the adaptive dissociation feature, HS/DAC coating not only significantly reduced thrombus adhesion by 84.6% in an acute canine model, but also exhibited antibacterial performances by the implantation of catheters in ex-vivo rabbit model (Fig. 4 and Supplementary Fig. 11 in revised manuscript and Supplementary Information), which are rarely reported.

5. The author first made bacteria adhere to the surface of the catheter, and then conducted blood tests to prove that bacterial adhesion can promote coagulation. It is suggested that the authors add the opposite test, exposing the material to blood and then conducting antibacterial tests, to show that the clots promote bacterial colonization or reduce the bactericidal properties of the material. Because the author mentioned in the preface that the antibacterial and anticoagulant are mutually reinforcing, the experiment in one direction is not enough to verify the author's view.

Our response: We have added the opposite experiments to investigate the antibacterial and anticoagulant of HS/DAC coating. The experimental process and discussion are listed in the following and also revised manuscript. Typically, bare films and HS/DAC coated films were immersed in blood for 1 h, and the blood was calcified with 0.045 mol/L calcium chloride. It can be clearly observed that there is no obvious blood clot on the surface of the coated film, while a significant thrombus has formed on the surface of the bare film. Then, samples were gently rinsed with saline and soaked in bacterial solution (10^6 CFU/mL) for 12 h. Finally, the samples were lightly washed with PBS buffer and ultrasonicated, and 200 μ L of solution was taken for colony counting. As shown in Figure R4, the number of bacteria on the surface of the bare films after blood treatment is 740 times higher than that of the HS/DAC coated group, and the bactericidal rate of the HS/DAC coating was still as high as 99.74%. In fibrinogen (Fg) adhesion experiments, more activated Fg was adhered on the surface of the bare film than the coated one, thereby promoting the formation of thrombus (Fig. 3b). Components of thrombus, such as fibrin, fibronectin and blood platelet, can act as anchors for bacteria adherence, which in turn enhances bacterial growth (*Science Translational Medicine*, 2012, 4, 153ra132; *Blood*, 2020, 136, 1).

Therefore, this opposite experiments also confirmed the stable antibacterial and anticoagulant of HS/DAC coating.

Figure R4 Agar plate colony counting assay of the blood pretreated bare films and the blood pretreated coated films, n=4. $***p < 0.001$.

6. In Figure D5, there is a spelling error, bacteria are written as Bactereria, and it appears twice.

Our response: The typo error of “Bactereria” in Fig 5d is corrected to be “Bacteria”. And we also fully checked the writing English in our manuscript.

Thank you again!

Reviewer #2:

In this work, authors present a coating for catheters, its synthesis and the advantages it presents compared to uncoated catheters, which include antimicrobial and antithrombotic activity, and study biocompatibility of these modified catheters. Moreover, authors test the coatings resistance and properties under various stress conditions, obtaining excellent results. The topic seems novel, and the new material appears to be an important solution to various health problems regarding complications due to catheter use in patients. Work is extensive and sound and most conclusions are well supported by it.

Paper could benefit from a proof reading by a native English speaker, as redaction is a bit unclear at times, especially in the introduction and results portions, and some colloquial language is used. Moreover, distribution of information throughout the paper is confusing: the results carry most of the discussion, whereas the discussion seems more like conclusions or a summary of the results. Discussion would be richer if some comparison with other types of coatings was presented. Some methods have text that I would categorize as introduction or discussion. Below, some comments and clarifications that could benefit the paper:

Our response: We sincerely thank Reviewer #2 for the evaluation of our manuscript and insightful comments. We have reorganized the results and discussion sections in revised manuscript. And we also fully checked the writing English in our manuscript.

A commercial infection-resistant CVC catheter (Arrowgard Blue, 16 Ga, Teleflex) was selected as control sample to compare the antibacterial and antithrombotic properties with our HS/DAC sample. For the antithrombotic experiments, blood from rabbit was drawn fresh into sodium citrate containing vacutainers (volume ratio of anticoagulant to blood = 1:9). CVC, (Arrowgard Blue)-CVC, and (HS/DAC)-CVC were cut into 3 cm pieces. All catheters had been previously sterilized and rinsed with saline solution thoroughly. Samples were implanted into silicone rubber catheters with an inner diameter of 3 mm, respectively and connected to a peristaltic pump. After that, 3 mL of citrated blood was recalcified using 0.045 M calcium chloride in saline (50 μ L/mL) and injected into the peristaltic pump to form a closed loop system, and incubated at 1 mL/min for 2 h. Samples were gently washed with PBS and used for weighing. As shown in Figure R5a, compared to commercial one, the thrombus rate of HS/DAC coated CVC decreased by 62.5%. This indicates that the HS/DAC coating has potential advantages in inhibiting thrombosis and avoiding side effects caused by

heparin injection methods in clinic.

For the antibacterial experiments, Bare CVC, (Arrowgard Blue)-CVC and (HS/DAC)-CVC were immersed in the bacterial solution (1 mL, 10^6 CFU/mL) for 12 h. Samples were lightly washed with PBS buffer solution and added with 6 mL PBS solution for ultrasound, and then 200 μ L of solution were taken for colony counting. As shown in Figure R5b, the bactericidal rate of commercial (Arrowgard Blue)-CVC and (HS/DAC)-CVC against *S. aureus* was more than 99%. Above all, the HS/DAC coated CVC exhibits better antithrombotic and antibacterial performances than commercial anti-infection CVC.

Figure R5 (a) Photographs of the thrombi on the surface of samples and cross-sectional observation of the samples. (b) Quantitative analysis of samples before and after circulation (n = 4). *** $p < 0.001$, ** $p < 0.01$.

(1) Introduction

a It is stated that “90% of patients require IV catheters”. Does this refer to cancer patients? Would need a reference.

Our response: We feel very sorry for this confused description! The 90% of patients who need IV catheters treatments are not merely refer to cancer patients. It means that about 90% of patients admitted in hospital require short- or long-term IV catheters to deliver medications, nutrient solutions, and blood. We have corrected this in the revised manuscript as follows: “Approximately 90% of patients require IV

catheters-mediated therapies, and more than 5 million central venous catheters (CVCs) are applied on the treatment of cancer in the USA (*Lancet Infect Dis*, 2016, 16, 241-250; *Journal of Clinical Oncology*, 2003, 21, 3665-3675).” And these corresponding works are cited in Ref 2 and 3.

b I’m unclear as to what “20-400 thousand of mortality” means. Is it number of people? In the world? In which period of time?

Our response: We have corrected the unclear expression in the revised manuscript as shown below: Although great efforts have been devoted to avoid bacterial infection, the mortality rate of CRBIs remains high at 12-25% occur annually in the United States (*The Lancet Infectious Diseases* 2016, 16, 241; *Current Infectious Disease Reports*, 2005, 7, 413). And these corresponding works are cited in Ref 3 and 12.

(2) Results

a The paper states “(HS/DAC)-TPU exhibited obvious inhibition of FXa and FIIa” and references a supplementary figure. It would be interesting if these results were further commented, either qualitatively or quantitatively.

Our response: Thank you for your suggest. We have moved the above mentioned results of the anti-FXa and anti-FIIa ability of the (HS/DAC)-TPU in the revised manuscript as shown in Figure R6a, b. The discussions about the quantitative analysis of HS activity in (HS/DAC)-TPU have been added in the revised manuscript and also as follows: “The conformation of antithrombin III (AT III) changes after binding to HS, which inhibits the activation of procoagulant factors FX and FII. Proteases FXa and FIIa are subsequently inactivated, thereby inhibiting coagulation. Compared with TPU and DAC-TPU, (HS/DAC)-TPU reduced the formation of FXa and FIIa by 95.8% and 94.3%, respectively (Fig. 3a,b).”

Figure R6 Evaluation of the anti-thrombotic ability of the HS/DAC coating *in vitro* and *in vivo*. (a) Anti-FXa and (b) anti-FIIa assays of the bioactivity of heparin on the HS/DAC-coated TPU (n = 3). (c) Amounts of BSA absorbed on bare and HS/DAC-coated TPU, as determined by a BCA protein assay kit (n = 9). (d) CLSM images of Fg adhesion, activation and leukocyte adhesion on the surface of bare catheters and HS/DAC-coated catheters. (e) Schematics of the antithrombotic mechanism of HS/DAC-coated catheters. (f) Digital and SEM images of the thrombus on the bare and HS/DAC-coated CVC catheters in the acute canine model *in vivo*. (g) Process by which the electrostatic interaction of the HS/DAC coating dissociates and

the results of WCA and ATR-FTIR spectra of bare and HS/DAC-coated TPU before and after treatment with PBS (n=6). *** $p < 0.001$, ns: no significant difference at $p > 0.05$.

b BCA assay and leukocyte adhesion have no methods or references.

Our response: The leukocyte adhesion experiment was added in the part “Inflammatory cells adhesion adsorption” in the revised Supporting material and following. The BSA adhesion experiment was as follows: “Samples were soaked in PBS solution of BSA (1 mg/mL) for 1 h at 37 °C. Then gently rinse three times with PBS to remove un-adsorbed proteins. The BCA protein kit was used to detect the amount of protein adsorbed on the sample surface. Sodium lauryl sulfate solution (SDS, 20 mg/mL) was added to each sample and shook for 2 h to detach the proteins on the surface. The solution was then supplemented with an equal amount of BCA reagent and incubated at 60 °C for 1 h. The absorbance of each sample was read and recorded at 560 nm (SYNERGY).”

c The colony count assay as explained (sonication of bacteria in catheter, suspension and distribution in an agar plate) gives a notion of how many bacteria remained adhered to the surface, not necessarily of the coating’s ability to inhibit or kill them. In the contact inhibition assay, both catheters (coated and uncoated) seem to have the same inhibition capacity, restricted only to bacteria directly in contact with the material. These two assays combined don’t seem enough to conclude that the coating has an antimicrobial ability in a contact-killing mode.

Our response: Thanks for your advice. The number of colonies is replaced by the bactericidal rate, i.e. the percentage of bacteria reduced. HS/DAC coated TPU surfaces show much better antibacterial rate of above 99.99% and 99.12% against the *S. aureus* and *E. coli* in comparison with bare TPU (Figure R7a, b). And other colony count results were also changed to bactericidal rate (Fig. 4a, c, i and j and Supplementary Fig. 17 and 18 in revised manuscript and Supplementary Information). To determine the antibacterial mode of our HS/DAC coating, the new inhibition zone experiments were redone with aim to obtain more clearly digital photos for result analysis. As shown in Figure R7c and Supplementary Fig. 15, it can be obviously observed that there is no colony in the part of the HS/DAC coated TPU in contact with the bacteria, and the coated film is still transparent. On the contrary, a large number of colonies appear at the bottom of the bare TPU. In addition, the diameters of inhibition zone are all 0 mm of the HS/DAC coated TPU, indicating that the DAC is anchored to the surface by covalent bonds without leaching. Above all, those above

results suggested that HS/DAC coatings given a better bactericidal performance through contact-killing mode.

Figure R7 a Colony count of the HS/DAC coatings to *S. aureus* and *E. coli*. b Antibacterial activity of the HS/DAC coatings against *S. aureus* and *E. coli*. c Zone of inhibition tests of bare and HS/DAC coated TPU films (n = 4). *** $p < 0.001$.

d Antibiofilm activity has no method per-se, it's briefly redacted in results.

SEM shows cell adhesion and morphology. When looking at *E. coli* in bare TPU, a biofilm seems to have formed, but for *S. aureus*, cells have adhered, but the presence of a biofilm is not clear in images. Therefore, this experiment may not lead to the conclusion that coated TPU is able to inhibit biofilm formation in comparison to uncoated one, at least for *S. aureus*. Later on, the authors refer to a visible biofilm formation in bare catheters and not in coated ones. Moreover, SEM images showed that coating changes bacterial morphology, which can affect biofilm formation. All of this information combined could lead to a sound theory of the coated material's ability to inhibit biofilm formation, but does not seem conclusive.

Our response: To testify the ability to inhibit biofilm formation, the anti-biofilm activity of the HS/DAC coated TPU further investigated by extending the co-incubation time of the sample with bacteria. Typically, TPU and HS/DAC coated TPU (1.0 cm × 0.8 cm) were incubated in Luria-Bertani (LB, 200 μ L) containing 10^6 CFU/mL of *S. aureus*. After the incubation at 37 °C for 12 h, the surfaces were gently rinsed with PBS. The medium was refreshed at 24 h. After 48 h, colony plating was performed. In addition, each sample was washed gently with PBS and fixed with 4% polyformaldehyde for 6 h. Then SEM was used to observe. As shown in Figure R8a and b, HS/DAC coated TPU films were mostly free from bacterial colonization and no trace of mature biofilm was found. Compared to the bare TPU, HS/DAC coating reduced the biofilm coverage by 94.7% at 48 h. In addition, the antibacterial

experiment of the catheter at 48 h also proved that the coating has the potential to inhibit biofilm formation (Figure R8c). The corresponding descriptions were marked in yellow in the revised manuscript and Supporting information.

Figure R8 Antibiofilm activity of the coating. (a) Agar plate colony counting assay and (b) SEM images of the bare TPU and (HS/DAC)-TPU incubated with bacteria for 48 h (n = 3). (c) Images of catheters and live bacteria adhering to the inner surfaces of the catheters and PVC tubes in flow conditions at different times. *** $p < 0.001$.

(3) Methods

a In the stability assay, triglyceride (30 v/v% in water) is mentioned as a solvent. I'm not sure it is, or if it can be dissolved in water.

Our response: We apologize for this confusion we caused by miswriting glycerol as triglyceride. Glycerin and water can dissolve each other in any proportion. Glycerol can increase the solubility of poorly soluble drugs.

b In Anti infection-related thrombus and catheter-related bacteremia of HS/DAC coating, the use of three rabbits is mentioned, but then four different groups appear.

Seems like each group had three rabbits, but redaction is unclear.

Our response: I am sorry for these unclear descriptions. The four different groups of samples were the bare catheter, the coated catheter, the bacterial infected bare catheter, and the coated catheter. All four groups were tested on the same rabbits to minimize the systemic errors. In order to avoid the influence of bacteria, we first conducted the experiment without bacteria. Four sets of parallel samples were tested. The corresponding descriptions were corrected in revised manuscript and also Supporting information.

c In antibacterial properties in vivo (supplementary), it's commented that "catheters were sterilized by immersing in 75% ethanol for 30 min". Ethanol is not a sterilizing agent, but rather a disinfecting one.

Our response: Reviewer is right! We have corrected the sentence as "Next, bare and HS/DAC coated indwelling catheters were disinfected with 75% ethanol for 30 min." in revised manuscript.

d Antibacterial assays have no positive controls reported.

Our response: Considering the potential application, all antibacterial tests in this work were performed according to the JIS Z 2801 standard. And there is no requirement for a positive control in this standard. Antibacterial activity value $R = \log(B/C)$. B and C represent the average number of alive bacteria adhered to the surface of samples without antibacterial properties and antibacterial samples after 24 h of bacterial inoculation, respectively.

e Clotting time method is explained in supplementary materials but has no results.

Our response: For the uncoated glass bottle, the mean occlusion time was 221 s, while the mean occlusion time for the modified glass bottle was more than 2 h as shown in Supplementary Fig. 10. This test indicated that the presence of heparin can lengthen the clotting time due to the inhibition of protease activity. These above results have been added in revised Supporting information. Including coating membrane antibacterial, coating catheter antibacterial, in vivo antibacterial, long-term antibacterial stability and biocompatibility

Thank you again!

Reviewer #3:

The manuscript describes an antithrombotic and antimicrobial coating made of heparin and a silane with a quarternary ammonium group. The coating binds firmly to various medical devices, such as catheters, after a hydroxyl-group forming activation step of the surface. The authors provide structural analysis of the coating and functional in vitro and animal tests.

The presentation of the study would need intensive restructuring. The authors performed and present multiple experiments, but do not introduce the principal setup and specific question of the experiment in the text. Multiple different analyses are presented in a single paragraph without any logical sectioning. Important information, such as the full name of DAC, a main coating component, is found only in the supporting information. The language is well understandable but would need careful revision. A more restrictive use of abbreviations would simplify reading. This type of presentation strongly impedes careful scientific review. The reviewer would be very willing to review a reorganized manuscript again. One conceptional problem can be indicated here: The antimicrobial action of quarternary ammonium salts and the anticoagulant activity of heparin are both closely linked to their charge. The complex formation in this coating neutralizes these charges. The authors phenomenologically demonstrate high (residual) activity. However, this should be rated to the activity of the native compounds.

Our response: Thank the reviewer for the valuable guidance. According to the suggestions of the reviewers, the purpose of the experimental design, the logic of the paragraphs, and the abbreviations of various materials were carefully corrected in revised manuscript and also Supporting information. For example, to demonstrate the facile manufacturing methods of the coating on various medical devices, the HS/DAC coating was utilized on various medical devices via staining method. In the part of “Antibacterial and biocompatibility properties of the HS/DAC coating *in vitro* and *in vivo*”, we divide the previous paragraph into five paragraphs. The verification was carried out from the aspects of the broad-spectrum antibacterial activity, antibacterial of the coated catheter, antibacterial *in vivo*, long-term antibacterial stability, and biocompatibility. The abbreviations of various materials and characteristics, such as DAC, CLSM, XPS, etc, were added in the revised manuscript.

Surely, the ratio of positive and negative charges and their bonding situations under physiological conditions are of great significance to the simultaneous antibacterial and

anticoagulant properties. To illustrate this salt-triggered adaptive dissociation mechanism, several spectroscopic characterizations and antibacterial and anticoagulant experiments have been carried out. As shown in Fig. 3g and Supplementary Fig. 14, the WCA of the (HS/DAC)-TPU decreased significantly after incubation with PBS solution. The band of carboxyl group shifted from 1609 to 1618 cm^{-1} , and the sulfonic acid group decreased and shifted significantly, attributing to the destruction of electrostatic interactions between the anion group and the quaternary amine group. XPS spectra show that the S 2p peaks are shifted from 167.9 and 168.95 eV for the HS/DAC coating to 168.15 and 169.40 eV for the salt treated HS/DAC coating, respectively. Additionally, the N 1S peak is slightly shifted from 402.05 to 402.15 eV after salt treatment, demonstrating the dissociation between HS and DAC. Our present results demonstrated that there above was not affect the antibacterial and antithrombus properties of the HS/DAC complex and its coating both *in vitro* and *in vivo*. To further investigate the correlation between charge ratio and bio-performances, DA/QAC complex with nearly neutralizing charges was synthesized as model to simulate the fully ions shielded situation. And the changes of antibacterial properties were studied by detecting the MIC of QAC, DA and DA/QAC, respectively. As shown in Figure R9, the calculated MIC value of DA/QAC was at the same level as that of free QAC. This confirmed that electrostatic bonding does not affect the biological properties of the material itself.

Therefore, we can make conclusion that the formed HS/DAC coating can take place the salt-triggered adaptive dissociation under the simulated physiological condition to expose the sulfonate and quaternary ammonium ions, benefitting to maintain the broad-spectrum antibacterial and antithrombus properties. This similar mechanism has also confirmed in previous works including the small molecular salt ions and polyelectrolyte complexes (for example, *Macromolecules* 2022, 55, 3, 978; *Phys. Rev. Lett.* 2010, 105, 208301; *Macromolecules* 2020, 53, 1, 102). These above descriptions were marked in yellow in the revised manuscript and Supporting information.

Figure R9 Molecular structures and *in vitro* antimicrobial properties of DA/QAC model complex with neutralizing charge.

Thank you again!

Reviewer #4:

This important work addresses the need for antithrombotic and antimicrobial catheters, by developing novel heparin-containing polyelectrolyte complexed coatings on polyurethane catheters. There are some important items that should be fixed prior to publishing this work.

1. Three different animal models were used in this work (rabbits, rats, and canines). The authors do not offer justification for the selection of these animal models. They also do not indicate the number of animals used (for the canine studies) or the justification for the number of animals used (e.g. power calculations) determining how the number of animals in each group was selected. Whether blinding and randomization were used for each study are also not reported. I cannot recommend publication of this work without sufficient justification. I recommend that the authors consult and reference standardized/recommended ethical guidelines for designing and reporting these experiments, such as the ARRIVE guidelines. (<https://doi.org/10.1371/journal.pbio.3000411>)

Our response: Thank the reviewer for the valuable guidance. According to the 3R principle "Reduction, Replacement, Refinement", mice are preferentially used as an evaluation of antibacterial performance. Since mice are too small to be suitable for antithrombotic performance *in vivo*, rabbits were selected as short-term arteriovenous circulation models to evaluate thrombotic performance. Subsequently, based on the results of rabbit experiments, beagle canine that was more in line with acute thrombosis was further selected as an animal model. The stability of the coating *in vivo* was studied by selecting a rat animal model with convenient operation.

Animal welfare is improved by minimizing the number of animals used within a reasonable design. The sample number of the mouse *in vivo* experiment was six. The sample number in the animal model experiment of rabbits was four. The sample number of canine animal model experiment was three. The sample number in the animal model experiment of rats was five.

All samples were randomly allocated into experimental groups. Single-blind methods were used for mouse and rat animal models, and double-blind methods were used for rabbit and beagle canine experiments.

2. The researchers conducted molecular dynamics simulations, but there is no discussion of how the results of these simulations impacted the study design or the interpretation of the results. This should be either made relevant to the reported work

or it should be removed from this report.

Our response: Considering that the molecular dynamics simulations is weakly relevant to the main content of this work, this part has been removed from the revised manuscript.

3. When reporting the catheter weight, following the rabbit circulation model, is this the total weight of the catheter, or the change in the weight associated with the thrombus formation (fig. S12). This should be clarified and interpreted. Also this should somehow be normalized to the contact area of the blood to the inside surface of the catheter. An absolute mass for the thrombus without some relative measure of the area or size over which the thrombus formed is not very meaningful.

Our response: The catheter weight following the rabbit circulation model is the change in the weight associated with the thrombus formation. In the image of the HS/DAC-coated catheter, a noticeable reduction was observed in thrombi and occlusion formation compared with that of the bare catheter, and the mass of the thrombus was reduced by 85%. (Supplementary Fig. 10 in revised Supporting information). We also agreed with the reviewer's suggestion to normalize the thrombus mass. The thrombus formed on the film or catheter with calculable area was normalized (Fig. 5b and Supplementary Fig. 10 and 13).

4. For the coating stability study using flow, the authors should calculate the shear rate at the surface, and justify the use of (Newtonian?) saline solution versus non-Newtonian blood (under the viscosity and shear rate ranges used). These values should also be compared to physiologically relevant ranges of shear rates encountered during catheter use.

Our response: Reviewer is right! The shear rate of 0.9% NaCl solution in flowing environment is 30 s^{-1} . It is significant to evaluate the stability of the coating in combination with the flow rate of human blood. However, it is difficult for us to select a specific shear rate for stability research due to the complexity of blood flow rates in different parts of the human body. Therefore, we investigated the long-term stability of the HS/DAC coating under different shear rates in artificial blood and under the short-term high speed water impact.

To demonstrate the long-term stability of the HS/DAC coating in blood, we tested the antithrombotic and antibacterial properties of the coating after 30 days of treatment in flowing artificial blood via a multichannel circulatory system (Figure R10a). HS/DAC

coated TPU catheter (ID 0.3 cm) were connect to the silicone tube. Then a peristaltic pump (LEAD FLUID, BT101 L) was utilized to connect the channel. Artificial blood was injected into the peristaltic pump to form a closed loop system. It is known that the nature of blood flow can be represented by a power law fluid (*Commun Nonlinear Sci Numer Simulat*, 2013, 18, 1970; *J. Non-Newtonian Fluid Mech.*, 2000, 94, 47). The shear rate of a non-Newtonian fluid can be calculated by the following formula, where n is the power-law index, Q is the flow rate, and R is the diameter of the conduit.

$$\gamma = \frac{3n + 1}{n} \times \frac{8Q}{\pi R^3}$$

Therefore, we investigated the stability of coatings with shear rates of 30 s^{-1} and 300 s^{-1} respectively. For the antithrombotic stability of coating, blood from rabbit was drawn fresh into sodium citrate containing vacutainers (volume ratio of anticoagulant to blood = 1:9). The catheter at 30 days were cut into 3 cm pieces. Samples were connected to the silicone tube. After that, 3 mL of citrated blood was recalcified by using 0.045 M calcium chloride in saline ($50 \mu\text{L}/\text{mL}$) and injected into the peristaltic pump to form a closed loop system, and incubated at $1 \text{ mL}/\text{min}$ for 1 h. Samples were gently washed with PBS and used for weighing. As shown in the Figure R10b, a large number of thrombus was formed in the lumen of the bare catheter, while no obvious thrombus was observed in the lumen of the coated one. Compared with the bare catheter, the thrombus of the HS/DAC coated catheter was reduced by 97.8%, 81.7% and 61.7% at the shear rate of 0 s^{-1} , 30 s^{-1} and 300 s^{-1} , respectively (Figure R10c).

To confirm the long-term antibacterial, the HS/DAC coated catheters at 30 days were soaked in the bacterial solution ($10^6 \text{ CFU}/\text{mL}$) for 12 h. Samples were lightly washed with PBS buffer solution and added with 8 mL PBS solution for ultrasound, and then $200 \mu\text{L}$ of solution were taken for colony counting. As shown in Figure R10d, e, the bactericidal rates of the coating at shear rates of 0 s^{-1} , 30 s^{-1} and 300 s^{-1} were 98.65%, 98.14% and 97%, respectively. Although the bactericidal rate decreased slightly, the coating still had good antibacterial performance at higher shear rate.

Additionally, the structure and performance stability of the coating were evaluated by impacting with water flow for 30 min. The velocity of the water flow was $50 \text{ mL}/\text{s}$, and the distance of the sample ($0.5 \times 0.5 \text{ cm}$) to the water source was 0.2 m. The staining and SEM images results suggested that the coatings were firmly fixed on the substrate in the water shearing environment (Figure R10f). As shown in the Figure R10g, no significant differences in the quality of thrombus were been observed in

comparison with that untreated coating. Meanwhile, the bactericidal rate of the coating remained about 99%.

These above results confirmed the long-term stability underflow of our HS/DAC coating, and the corresponding descriptions were added in the revised manuscript and Supporting information marked in yellow.

Figure R10 **a** The multi-channel circulation model of long-term stability underflow. **b**. Thrombosis on the surface of the coating after flowing at 0 s⁻¹, 30 s⁻¹ and 300 s⁻¹ for 30 days. **c** The corresponding thrombus mass changes. **d** The colony count of HS/DAC coating after flowing at 0 s⁻¹, 30 s⁻¹ and 300 s⁻¹ for 30 days. **e** The corresponding antibacterial rate. **f** The staining and SEM images of the HS/DAC coating with high speed water impact. **g** The corresponding thrombus mass changes. **h** The corresponding antibacterial rate.

5. The entire manuscript should be edited for English language, grammar, and usage. Some errors obscure the meaning.

Our response: We have revised the language and grammar of the entire manuscript.

Thank you again!

REVIEWER COMMENTS

Reviewer #1 (Remarks to the Author):

The author answered my questions sufficiently, and I think the revised manuscript is ready for publication in the NC Journal.

Reviewer #2 (Remarks to the Author):

In the revised version of the manuscript, authors either corrected or clarified each of the points mentioned in my previous review. Moreover, the revised version is more clearly redacted and organized, which leads to a better understanding of the work.

Reviewer #3 (Remarks to the Author):

Review to Ms Review NCOMMS-22-43549B

Lin Liu, et al., Heparin-Network-Mediated Long-lasting Coatings on Intravascular Catheters for Adaptive Antithrombosis and Antibacterial Infection

General

The authors substantially revised their study's presentation, which can now be followed easily. This is appreciated.

The coating in the study consists of the anticoagulant heparin (HS) and a silane with an antimicrobial quaternary ammonium group. It is stabilized by organosilane groups and salt bridges between its components. These compounds are soluble in organic solvents and can be applied as coatings on various substrate materials, where the silane groups of the coating support binding to OH-bearing substrates. In the high-ionic biological environment, the salt-bridges between heparin and the quaternary ammonium groups open ("adaptive dissociation in salt solution"), and the two compounds gain their biological activity again.

The concept is well elaborated but does not appear highly innovative. Multiple analysis of the biological performance have been made, but there are some misleading comparisons (see below) and mistakes in the presentation of the anticoagulant properties.

The discussion section is actually a summary and conclusion. It does not provide mechanistic interpretations of the observations or bring this study in a context with data of other groups.

Major

Line 159: the statement that heparin functions as a serine protease inhibitor is wrong in this phrasing. Heparin catalyzes the activity of the serpin antithrombin III.

Line 160 the heparin-Antithrombin complex does not inhibit the conversion of FX and prothrombin to

FXa and thrombin, respectively (“activation”), but (irreversibly) inhibits the activated proteases. Correspondingly (line 165), not the formation of FXa and FIIa were reduced, but their activity (the experimental section states that active FXa and FIIa were added to the samples). The corresponding experimental section (SI) “Bioactivity of the HS/DAC coating” is not precise: “...films were incubated” should indicate the medium.

Lines 169/170: fibrinogen activation to fibrin is the final step of the coagulation cascade and not its initiator. If the authors refer to platelet activation and aggregation by denatured fibrinogen, they should express more clearly.

Line 173f and line 215: the text states about activated fibrinogen (coagulation factor I), but figure 3b states thrombin (coagulation factor IIa). The experimental section indicates the analysis of fibrin/denatured fibrinogen. The authors should differentiate between denatured fibrinogen and a fibrin mesh e.g. by providing SEM-images.

Figure 3c: the panel (axis title) should provide information on what is indicated. – Albumin adsorption (Fig 3c) was determined from immersion in buffered albumin solution, whereas fibrin(ogen) adsorption (Fig 3b) was obtained from a whole blood clotting experiment. This experimental difference must be indicated in the caption and in the main text. The parallel presentation of these results causes misleading conclusions.

Line 275ff: the biocompatibility (actually cytotoxicity) test was performed with extracts of the materials and not as direct contact test. This needs to be indicated in the text. Direct contact tests also appear necessary in this context. As the antimicrobial tests show contact kill by the surface, such direct toxicity must be excluded for eucaryotic cells.

Line 178 and 327: Why did authors check for monocyte and lymphocyte (T-cell) adhesion and not granulocyte adhesion? Granulocytes are most abundant and involved in the primary defense. Lymphocytes do not actively adhere to foreign surfaces but mainly get passively entrapped in a fibrin clot.

The authors should analyze a heparin release from their coating.

Minor

In this revised version, still a more conservative use of abbreviations in the abstract and introduction would be recommended. Many abbreviations are not used further or appear only rarely in the text. The abbreviation HS for heparin is not common in the community.

Lines 200ff: the experimental setting of these results should be described briefly. As the lines before describe studies in buffer solution, the presentation to blood clotting is not obvious.

Subsections of long paragraphs that present multiple different results could facilitate reading further (e.g. lines 190-211, 223-239, 302-331 and others)

Lines 65, 69: In the introduction, the terms “long term” “long-lasting” should be specified a bit better. The term “undflow treatment” is misleading and might be replaced by “flow incubation” or “shear incubation”.

Line 78, 204f: The terms “adaptive dissociation” and “self adaptivity in salt solution” would need more introduction.

Line 141: The bending radius in the stability test should be indicated.

Reviewer #4 (Remarks to the Author):

This is a very thorough investigation of the proposed coatings for catheters. There is a need for catheter surfaces that have multiple functions, including inhibiting infection and clotting, as proposed here. Since the authors have conducted a lot of experiments particularly to determine biological outcomes, it is difficult to follow the rationale and hypothesis being tested in each experiment. These could be more clearly stated, for example, separate antibacterial activity assays are conducted in vitro and in vivo, including two different models (one of inoculated samples implanted into mice, and another of samples implanted into rats). Why are both studies necessary and how do they answer different important scientific questions?

While the authors have included a thorough response to the reviewers comments, not all of these responses are reflected in the revised submission:

1. Authors have not sufficiently responded to Reviewer 2 comment #1. In their responses to the comments they clarify that, "90% of patients admitted in hospital require short- or long-term IV catheters to deliver medications, nutrient solutions, and blood." However, the revised text simply reads, "Approximately 90% of patients require IV catheter-mediated therapies, and more than 33 million central venous catheters (CVCs) are applied to treat cancer in the USA^{2, 3}." The text does not qualify these as "patients admitted in hospital." Moreover the text may be misinterpreted to understand this as 90% of cancer patients. "Patients" is too broad.
2. Comment #3 from reviewer 2 has not been addressed. "triglyceride (30 v/v% in water)" is still mentioned on line 392.
3. The numbers of animals used is still not clearly reported. The number of mice (9) used in the antibacterial studies is reported in the supporting information. However, I do not see any reference to the numbers of dogs (in the thrombogenicity study), rats (clotting time, fibrinogen adsorption and activation studies, antithrombotic stability studies, and antibacterial stability study), or mice (biocompatibility study) in the text of the methods, results, or supporting information. Four rabbits are indicated in line 424, and these were divided into four groups (line 431). Is this one rabbit in each of four groups (four total) or four rabbits in each of four groups (16 total)? The authors have used a large number of animal models in this study. For each experiment, animal use must be justified and overseen appropriately, in a way that the reader can easily find this information.
4. The authors have not fixed the error indicated by reviewer #2 (comment 2d). They still refer to 75% ethanol for sterilization on line 229 of the supporting information.

A few other corrections should be made:

5. Figure 3b and supplementary figure 12(b) the vertical axis is labeled "Ila (arb. U at 405 nm)." I think that this is supposed to be "FXIIa (arb. U at 405 nm)"
6. In the caption for figure 4b, the images are identified as from a coculture. Is this a coculture of two different cell types, or were the E. coli and S. aureus added cultured on separate samples? I think coculture is the incorrect term to describe these experiments, but perhaps I do not understand the experiment correctly.
7. Page 16, line 321 mentions "these four groups." In the preceding text describing the results of this experiment there is no description of what the four groups are. These four groups should be briefly described for the reader here, since the methods are included in the paper after the results section. This is just one example of how the results are difficult to interpret. More context and rationale for each experiment (e.g., exactly what hypothesis is being tested in each experiment, why were the conditions

used chosen, and how should the outcome of each experiment be interpreted in light of the other experiments) should be included in the results section.

8. What species and cell type was used for the cell viability studies, and what was the rationale used for this selection?

9. The sentence beginning "DAC ..." on lines 95-97 is grammatically incorrect and needs to be clarified.

Point-to-point response:

Reviewer #1

The author answered my questions sufficiently, and I think the revised manuscript is ready for publication in the NC Journal.

Our response: We are deeply gratefully for the review's positive feedback on our work.

Reviewer #2:

In the revised version of the manuscript, authors either corrected or clarified each of the points mentioned in my previous review. Moreover, the revised version is more clearly redacted and organized, which leads to a better understanding of the work.

Our response: We sincerely appreciate the reviewer's positive feedback on our work.

Reviewer #3:

General

The authors substantially revised their study's presentation, which can now be followed easily. This is appreciated.

The coating in the study consists of the anticoagulant heparin (HS) and a silane with an antimicrobial quaternary ammonium group. It is stabilized by organosilane groups and salt bridges between its components. These compounds are soluble in organic solvents and can be applied as coatings on various substrate materials, where the silane groups of the coating support binding to OH-bearing substrates. In the high-ionic biological environment, the salt-bridges between heparin and the quaternary ammonium groups open ("adaptive dissociation in salt solution"), and the two compounds gain their biological activity again.

The concept is well elaborated but does not appear highly innovative. Multiple analysis of the biological performance have been made, but there are some misleading comparisons (see below) and mistakes in the presentation of the anticoagulant properties.

The discussion section is actually a summary and conclusion. It does not provide mechanistic interpretations of the observations or bring this study in a context with data of other groups.

Our response: We greatly appreciate the reviewer's professional comments, which are of great help to our work.

1. Generally, whether from the application potential or good biological performance, the concept of "adaptive dissociation in salt solution" proposed by this work is of great significance for medical devices to cope with multiple complications. Due to the complexity of blood-contact medical devices and the interconnectedness of complications, there is currently no commercially efficient methods to address both catheter-related bloodstream infections and catheter-related thrombosis. First of all, the electrostatic assembly between HS and DAC is conducive to the efficient preparation of coating agents. The smart HS/DAC coating then adaptively dissociated in salt solution, which simultaneously reduced thrombosis adhesion and prevented biofilm formation.

2. We have modified the misleading expressions in revised manuscript and Supporting Information. For example, "HS catalyzes the activity of the serpin antithrombin III (AT III) and forms hydration layers to obtain antithrombin activity" in line 169. "Anticoagulant activity of the HS/DAC coating" in line 172, 248. "Fibrinogen (Fg) activation to fibrin is the final step of the coagulation cascade, in which the hydrolysis of thrombin plays a crucial role." in line 178. "Adherent Fg or platelets can promote the adhesion of leukocyte. Leukocyte, in turn, enhance the activation of local platelets and participate in the formation of thrombus" in line 192.

3. We have corrected the presentation of the anticoagulant properties in revised manuscript. For example, "Compared with TPU and DAC-TPU, (HS/DAC)-TPU reduced the absorbance of FXa and FIIa by 95.8% and 94.3%, respectively (Fig. 3a, b), indicating that (HS/DAC)-TPU significantly inhibited the activities of FXa and FIIa." in line 174.

4. The discussion section has been rewritten in revised manuscript, including the mechanistic explanation of the experimental results and the advantages compared with other work. For example, "Several antithrombotic or antibacterial strategies, classifying into noncovalent coating and covalent grafting, have been used to modify cardiovascular catheters. However, their practical applications are limited due to time-consuming construction strategies and poor long-term stability under complex physiological environments. In contrast to those methods mentioned above, our HS/DAC coatings may provide an available resolution for this issue."; "Furthermore, commercial functional catheters usually have a single bioactivity, such as medical catheters loaded with antibiotics for antibacteria or modified with heparin sodium for

antithrombotic usage. Therefore, it is difficult to simultaneously solve bacterial infections and thrombosis. According to Debye-Hückel theory, the electrolyte within physiological environments could weaken the electrostatic dipole-dipole interactions between HS and DAC, leading to the adaptive dissociation feature to meet the requirements of both antithrombotic and antibacterial capabilities simultaneously.”

“Compared with commercially available anti-infective CVC catheters (Arrowg+ard Blue), HS/DAC coated CVC catheters have comparable antibacterial properties, but the amount of blood clot is reduced by 62.5%.”

Major

1. Line 159: the statement that heparin functions as a serine protease inhibitor is wrong in this phrasing. Heparin catalyzes the activity of the serpin antithrombin III. Line 160 the heparin-Antithrombin complex does not inhibit the conversion of FX and prothrombin to FXa and thrombin, respectively (“activation”), but (irreversibly) inhibits the activated proteases. Correspondingly (line 165), not the formation of FXa and FIIa were reduced, but their activity (the experimental section states that active FXa and FIIa were added to the samples). The corresponding experimental section (SI) “Bioactivity of the HS/DAC coating” is not precise: “...films were incubated” should indicate the medium.

Reviewer is right! We have modified the inappropriate expression as follows:

(1) We have corrected the sentence as “As previously demonstrated, HS catalyzes the activity of the serpin antithrombin III (AT III) and forms hydration layers to obtain antithrombin activity. The conformation of AT III changes after binding to HS, which can inhibit the activation of coagulation factor Xa and thrombin (IIa), thereby preventing the occurrence of coagulation.” in revised manuscript.

(2) We have corrected the expression as “Compared with TPU and DAC-TPU, (HS/DAC)-TPU reduced the absorbance of FXa and FIIa by 95.8% and 94.3%, respectively (Fig. 3a, b), indicating that (HS/DAC)-TPU significantly inhibited the activities of FX and FII” in revised manuscript.

(3) We have corrected the “Bioactivity of the HS/DAC coating” as “Anticoagulant activity of the HS/DAC coating” in revised manuscript and Supporting Information.

(4) The medium is PBS buffer solution in order to remove residual impurities on the sample surface. We have supplemented the sentence as “First, bare TPU films and HS/DAC-coated TPU films were incubated in PBS for 2 min at 37 °C” in revised Supporting Information.

2. Lines 169/170: fibrinogen activation to fibrin is the final step of the coagulation cascade and not its initiator. If the authors refer to platelet activation and aggregation by denatured fibrinogen, they should express more clearly.

Our response: We have corrected the sentence as “Fibrinogen (Fg) activation to fibrin is the final step of the coagulation cascade, in which the hydrolysis of thrombin plays a crucial role.” in revised manuscript. This sentence expresses the activated fibrinogen, not the deformed fibrinogen.

3. Line 173f and line 215: the text states about activated fibrinogen (coagulation factor I), but figure 3b states thrombin (coagulation factor IIa). The experimental section indicates the analysis of fibrin/denatured fibrinogen. The authors should differentiate between denatured fibrinogen and a fibrin mesh e.g. by providing SEM-images.

Our response: We are very sorry that our carelessness led to the above misunderstanding. The result of activated fibrinogen is shown in Figure 3d, not Figure 3c.

The experimental section in Figure 3d indicates the analysis of fibrinogen adsorption and activation, not the fibrin/denatured fibrinogen. The blood circulation system *in vitro* was used to explore Fg activation on surfaces. Conversion of fibrinogen to fibrin is triggered by thrombin, which cleaves fibrinopeptides A and B from alpha and beta chains, and thus exposes the N-terminal polymerization sites responsible for the formation of the soft clot. The soft clot is converted into the hard clot by factor XIIIa which catalyzes the epsilon-(gamma-glutamyl) lysine cross-linking between gamma (γ) chains (stronger) and between alpha chains (weaker) of different monomers. Hence, the effect of HS/DAC coating on Fg activation was investigated by assessing the exposure of the γ chain via the immunofluorescence staining method, which has also been reported in other literature (Langmuir 2017, 33, 10402). As shown in Figure 3d, HS/DAC coated catheters expose significantly less γ chains compared to unmodified catheters. In summary, HS/DAC coatings avoid Fg activation and conformational changes by inhibiting thrombin and other coagulation cascade proteases. It has been reported that the denaturation of fibrinogen is mainly related to changes in the external environment, such as organic reagents, acidic solutions, high temperatures, etc (Macromol. Biosci. 2021, 21, 2000412; Acta Biomater. 2017, 54, 164; Biomacromolecules 2008, 9, 3258; Colloids Surf., B 2018, 167, 370).

4. Figure 3c: the panel (axis title) should provide information on what is indicated. – Albumin adsorption (Fig 3c) was determined from immersion in buffered albumin solution, whereas fibrin(ogen) adsorption (Fig 3b) was obtained from a whole blood clotting experiment. This experimental difference must be indicated in the caption and in the main text. The parallel presentation of these results causes misleading conclusions.

Our response: The experimental difference has supplemented in the caption and in the main text. For example, “c Amounts of BSA absorbed on bare and HS/DAC-coated TPU. Samples were immersed in buffered albumin solution and determined by a BCA protein assay kit (n = 9). d CLSM images of Fg adhesion, activation and leukocyte adhesion on the surface of bare catheters and HS/DAC-coated catheters (n = 3). Adhesion of Fg on the surface of the catheter was performed in a whole blood solution containing FITC-Fg. Circulating whole blood was used to detect the activation of Fg and adhesion of leukocyte” were modified in the caption of Fig. 3. “Confocal laser scanning microscopy (CLSM) images indicated that activated Fg on the surface of the HS/DAC-coated catheter circulating in whole blood was less adherent than the bare catheter, and negligible differences were observed in the amount of bovine serum albumin (BSA) by bicinchoninic acid (BCA) assay in the buffered albumin solution (Fig. 3c, d)” were indicated in revised manuscript.

5. Line 275ff: the biocompatibility (actually cytotoxicity) test was performed with extracts of the materials and not as direct contact test. This needs to be indicated in the text. Direct contact tests also appear necessary in this context. As the antimicrobial tests show contact kill by the surface, such direct toxicity must be excluded for eucaryotic cells.

Our response: In addition to the extraction method, the cytotoxicity of the HS/DAC coating was also investigated by contact method. The results of cell activity showed that HS/DAC coating also had good cytocompatibility. The experiments, results and discussion are as follows:

Experiments: Cells were prepared as previously described¹. The cytocompatibility was investigated by extraction method and contact method respectively. First, extraction method: Bare TPU, (HS/QAC)-TPU and (HS/DAC)-TPU were immersed in DMEM containing bovine serum (1%) for 24 h. After that, the extract of the sample is co-incubated with the seeded cells. After 24 h, add CCK-8 and place for 2 h to test

the absorbance at 450 nm. The negative control was cells without extract and the positive control is a blank sample without cells. Secondly, contact method: Bare TPU, (HS/QAC)-TPU and (HS/DAC)-TPU were incubated with the inoculated cells for 24 h, respectively. CCK-8 solution was then added for testing as described above. The preparation of negative and positive control samples was the same as above.

Results and discussion: Biocompatibility is a fundamental requirement for biomedical materials, and we first performed a Cell Counting Kit-8 (CCK-8) test to determine the cytotoxicity of HS/DAC to cell viability. As shown in Figure 19a, cell viability of (HS/DAC)-TPU was greater than 90% regardless of whether the coating was in direct contact with cells or extract in contact with cells. However, (HS/QAC)-TPU control has significantly reduced cell viability, especially as low as 5% when in direct contact with cells. Cytotoxicity experiments have shown that the excellent stability of the crosslinked HS/DAC coating can improve cell compatibility.

Supplementary Fig. 19 (a) Cell viability of bare, HS/QAC and HS/DAC-coated TPU via extraction mode (n = 8) and contact mode (n = 3-4).

6. Line 178 and 327: Why did authors check for monocyte and lymphocyte (T-cell) adhesion and not granulocyte adhesion? Granulocytes are most abundant and involved in the primary defense. Lymphocytes do not actively adhere to foreign surfaces but mainly get passively entrapped in a fibrin clot.

Our response: (1) Activated platelets combine with P-selectin glycoprotein ligand 1 (PSGL-1) on monocytes to form monocyte-platelet aggregates, which play an important role in promoting thrombosis and inflammatory response, and serve as a bridge between inflammation and thrombosis (Cell Mol Immunol., 2015, 12, 435; Sci. Adv., 2020, 6, eaaz1580). Similarly, lymphocyte-platelet aggregates are also formed between lymphocytes and platelets (Ann Lab Med., 2006, 26, 323; Thromb Haemost., 2019, 119, 821; Anaesthesia, 2003, 58, 312). Therefore, we detected monocytes and

lymphocytes adhering to the surface of the bare catheter and HS/DAC coated catheter after blood circulation via immunofluorescence, which has been reported in many reports (*Sci Transl Med.*, 2012, 4, 153ra132; *Clin Epigenetics.*, 2020, 12, 66; *Inflamm. Res.*, 2022, 71, 81; *Front. Immunol.* 13:941333). After whole blood circulation, clots formed on the surface of the bare catheter, while there were no obvious clots on the surface of the HS/DAC coated catheter. As shown in Figure 3d, a large number of monocytes and lymphocytes are attached to the surface of the bare catheter, which may be due to the activation of coagulation cascade reactions on the surface of the bare catheter. Prokaryotic cells and lymphocytes were passively trapped in fibrin clots. (2) Granulocyte does play an important role in the immune response. Granulocyte cells include neutrophils, eosinophils, basophils, and mast cells. At present, the main characterization method for detecting granulocytes is flow cytometry, which uses monoclonal antibodies labeled with different fluorescent substances to interact with the tested components, and then detects the tested cells using flow cytometry. The samples used for flow cytometry testing are usually in a flowable state, such as cell suspensions. In this work, it is difficult to perform flow cytometry detection due to the fact that only the surface of the bare catheter forms blood clot after the whole blood circulation, while the surface of the coated catheter does not. There are few reports on using immunofluorescence staining to detect granulocytes, and suitable primary and secondary antibodies cannot be obtained through research.

7. The authors should analyze a heparin release from their coating.

Our response: The formation of a cross-linked network between the HS/DAC coating and the substrate contributed to the good robustness for coatings. It is speculated that HS/DAC coatings are difficult to release HS. To verify whether the coating will release HS, the following experiments were conducted. The HS/DAC coated-TPU and Bare TPU (3 cm²) were soaked in PBS solution (3 mL, 37 °C) for 7 days. The content of element S in the extract was detected by Inductive Coupled Plasma Emission Spectrometer (ICP). As shown in Figure R1, only 2.3 ppm of S element was detected in the extract of the HS/DAC coating, and there was no significant difference compared to the bare TPU. This result demonstrated that the robust HS/DAC coating release little or no HS.

Figure R1 The content of S element in the extraction solution of the bare TPU and HS/DAC coated TPU (n = 3).

Minor

1. In this revised version, still a more conservative use of abbreviations in the abstract and introduction would be recommended. Many abbreviations are not used further or appear only rarely in the text. The abbreviation HS for heparin is not common in the community.

Our response:

(1) We have checked abbreviations in the article as shown in Table S1 except for the labeling of the sample and characterization. Abbreviations that appear only 1-3 times are deleted. For example, DVT, AT III and TT.

Table S1 Statistics of abbreviations used in the manuscript

full title	catheter-related bloodstream infection	catheter-related thrombosis	Intravascular	central venous catheter	deep vein thrombosis	antithrombin III	thrombin time
abbreviation	CRBI	CRT	IV	CVC	DVT	AT III	TT
usage amount	16	16	10	10	2	1	1

(2) HS is an abbreviation of heparin sodium and is commonly used in various literature (Angew. Chem. Int. Ed., 2018, 57, 15738; Adv. Mater., 2018, 30, 1706924). We have corrected the full name of HS to heparin sodium in revised manuscript.

1. Lines 200ff: the experimental setting of these results should be described briefly. As the lines before describe studies in buffer solution, the presentation to blood clotting is not obvious.

Our response: The experimental setting of these results was described briefly in

revised manuscript. For example, “Firstly, the anticoagulant function of HS/DAC coating was explored by thrombin time in vitro. HS/DAC-coated glass bottles have the longest clotting time in recalcified blood, indicating the potential of the coating for anticoagulant vascular applications (Supplementary Fig. 9a, b). More surprisingly, compared to commercial infection-resistant CVC catheter (Arrowgard Blue), the thrombus mass of HS/DAC coated CVC decreased by 62.5% (Supplementary Fig. 9c) through a closed circular tube loop of whole blood. This indicates that the HS/DAC coating has potential advantages in inhibiting thrombosis and avoiding side effects caused by heparin injection methods in clinical practice.”; “Therefore, the antithrombotic stability HS/DAC-coated TPU and its HS/QAC control was initially investigated via chromogenic assays of anti-FXa and anti-FIIa after incubated in 0.9% NaCl solution of 30 days.”;

2. Subsections of long paragraphs that present multiple different results could facilitate reading further (e.g. lines 190-211, 223-239, 302-331 and others)

Our response: Thanks to the reviewer for your suggestion, we have segmented long paragraphs in the revised manuscript. For example, lines 169-177, 178-191, 192-200, 201-209, 210-215, 216-221, 222-235.

3. Lines 65, 69: In the introduction, the terms “long term” “long-lasting” should be specified a bit better.

Our response: According to the reviewer's suggestion, the terms “long term” “long-lasting” was specified in revised manuscript. For example, “Hence, a robust (i.e., displaying long-term stability for up to 30 days in diverse environments), modular (structural and compositional flexibility), and adaptable (stimulus-responsive) coatings should be developed to obtain functional catheters with both antibacterial and antithrombotic properties to simultaneously treat CRBIs and CRT”. “Herein, we report a concise and straightforward route to construct a long-lasting (30 days) and adaptable coating that combines the high antithrombin ability of HS and excellent antibacterial feature of DAC to synergistically treat CRBIs and CRT”.

4. The term “underflow treatment” is misleading and might be replaced by “flow incubation” or “shear incubation”.

Our response: The reviewer's suggestion is reasonable. The “underflow treatment” or “underflow” was replaced by “flow incubation” or “flow” in revised manuscript. For

example, “Moreover, the remaining long-lasting anti-thrombotic stabilities of HS/DAC-coated TPU were also determined in flowing artificial blood and *in vivo*. No obvious thrombus was observed on the surface of the HS/DAC coating after flow incubation”.

5. Line 78, 204f: The terms “adaptive dissociation” and “self adaptivity in salt solution” would need more introduction.

Our response: According to the reviewer's suggestion, the terms "adaptive dissociation" and "self adaptivity in salt solution" were given more introduction in revised manuscript. For example, “Meanwhile, salts could weaken the electrostatic dipole–dipole interaction of intrachain. More functional groups in HS/DAC, such as sulfamate, sulfonate and carboxyl groups of HS and quaternary ammonium groups of DAC was exposed in electrolyte solution. The exposed hydrophilic groups were more conducive to reducing the thrombus adhesion rate (84.6%) in acute canine models and also exhibited a broad spectrum of antibacterial performances against *Staphylococcus aureus* (*S. aureus*) and *Escherichia coli* (*E. coli*).” “We speculate that the antithrombotic function of the HS/DAC coating originates from its self-adaptivity in salt solution. After the dissociation of HS/DAC fixed on the surface, more functional groups such as sulfamate, sulfonate, and carboxyl groups of HS was exposed, which is more conducive to the realization of antithrombotic function”.

6. Line 141: The bending radius in the stability test should be indicated.

Our response: The section of “The mechanical stability of the coating” in the part Materials and Methods, it was mentioned that “(HS/QAC)-TPU and (HS/DAC)-TPU tubes were simultaneously manually bent to 180° for 500 times”. Furthermore, the bending radius was also indicated in the part of Results as follows: “After bent to 180° for 500 times, (HS/DAC)-TPU showed little delamination or cracking”.

Reviewer #4:

This is a very thorough investigation of the proposed coatings for catheters. There is a need for catheter surfaces that have multiple functions, including inhibiting infection and clotting, as proposed here. Since the authors have conducted a lot of experiments particularly to determine biological outcomes, it is difficult to follow the rationale and hypothesis being tested in each experiment. These could be more clearly stated, for example, separate antibacterial activity assays are conducted *in vitro* and *in vivo*,

including two different models (one of inoculated samples implanted into mice, and another of samples implanted into rats). Why are both studies necessary and how do they answer different important scientific questions? While the authors have included a thorough response to the reviewers comments, not all of these responses are reflected in the revised submission.

Thank the reviewer for the valuable guidance.

Biological experiments follow the principle of exploring *in vitro* and then in animals. Firstly, the anticoagulant and antithrombotic properties of the coating were investigated *in vitro* to preliminarily determine the feasibility of the scheme. As shown in Figure 3a-d, Figure 4 a-e, Supplementary Fig. 8 and Fig. 15-18, the HS/DAC coating exhibited good anticoagulant and antibacterial properties. Therefore, we have conducted *in vivo* experiments that are closer to practical applications. According to the 3R principle "Reduction, Replacement, Refinement", mice are preferentially used as an evaluation of antibacterial performance. A mouse subcutaneous implantation model of bacterial infection was designed. HS/DAC coated catheters have shown great potential in preventing catheter-related bacterial infections, both in quantitative bacterial statistics and histopathological analysis Figure 4f. The implantation time of the catheter varies from a few days to a few weeks, so it is necessary to further explore the long-term stability of the coating *in vivo*. In order to reduce individual differences and compare the long-term antibacterial and antithrombotic stability of HS/DAC coated films, a rat subcutaneous implantation model was selected to facilitate the simultaneous implantation of several samples with larger size. After 30 days of implantation, the HS/DAC coated films can still kill 88% of bacteria and reduce thrombosis by 71% (Figure. 4j and Supplementary Fig. 13e). Since mice and rats are too small to be suitable for antithrombotic performance *in vivo*, rabbits were selected as short-term arteriovenous circulation models to evaluate thrombotic performance. Compared to bare catheters, HS/DAC coated catheters significantly reduce thrombosis and occlusion formation (Supplementary Fig. 10). Subsequently, beagle canine that was more in line with acute thrombosis was further selected as an animal model. After 24 h of implantation, few thrombi could be observed on HS/DAC-coated CVCs, but substantial visual thrombi were observed on the tip, side hole, and lumen of bare CVCs, which could cause CVCs to fail. Compared with commercial bare CVC, the coating reduced the mass of blood clots by 84.6% (Figure. 3f and Supplementary Fig. 11b).

Careful revisions were made to the errors raised by the reviewers in the revised article,

such as the confusing sentence in the part of Introduction.

1. Authors have not sufficiently responded to Reviewer 2 comment #1. In their responses to the comments they clarify that, “90% of patients admitted in hospital require short- or long-term IV catheters to deliver medications, nutrient solutions, and blood.” However, the revised text simply reads, “Approximately 90% of patients require IV catheter-mediated therapies, and more than 5 million central venous catheters (CVCs) are applied to treat cancer in the USA^{2, 3}.” The text does not qualify these as “patients admitted in hospital.” Moreover the text may be misinterpreted to understand this as 90% of cancer patients. “Patients” is too broad.

Our response: I am sorry for the unclear descriptions. The sentence was corrected as “Approximately 90% of patients admitted in hospital require short- or long-term IV catheters to deliver medications, nutrient solutions, and blood². More than 5 million central venous catheters (CVCs) are applied to treat cancer in the USA^{3, 4}” in the revised manuscript.

2. Comment #3 from reviewer 2 has not been addressed. “triglyceride (30 v/v% in water)” is still mentioned on line 392.

Our response: Thank you very much for your seriousness, and we apologize for our carelessness. The term of was corrected as “glycerol (30 v/v% in water)” in the revised manuscript.

3. The numbers of animals used is still not clearly reported. The number of mice (9) used in the antibacterial studies is reported in the supporting information. However, I do not see any reference to the numbers of dogs (in the thrombogenicity study), rats (clotting time, fibrinogen adsorption and activation studies, antithrombotic stability studies, and antibacterial stability study), or mice (biocompatibility study) in the text of the methods, results, or supporting information. Four rabbits are indicated in line 424, and these were divided into four groups (line 431). Is this one rabbit in each of four groups (four total) or four rabbits in each of four groups (16 total)? The authors have used a large number of animal models in this study. For each experiment, animal use must be justified and overseen appropriately, in a way that the reader can easily find this information.

Our response: We apologize for our carelessness. The number of animals used for testing in the article was shown in Table 2. Clotting time (n = 3), fibrinogen

adsorption (n = 3) and activation (n = 3), inflammatory cells adhesion (n = 3) studies were tested *in vitro*, the whole blood was collected from the same rat. The number of samples for each experiment was indicated in the captions.

Table 2 Animal testing and the number of used animals

Animal experiment	Number (n)
Mice used in antibacterial study	9
Mice used in biocompatibility study	6
Rats used in antithrombotic stability study	4
Rats used in antibacterial stability study	4
Rabbits used in antithrombotic properties	3
Rabbits used in bacteremia study	4
Dogs used in the thrombogenicity study	3

I am sorry for the unclear descriptions of the rabbits used for bacteremia research. A total of four rabbits were used to test the potential of HS/DAC coated catheters for bacteremia resistance. The samples used for testing were bare catheter, HS/DAC coated catheter, bare catheter inoculated with bacteria, and HS/DAC coated catheter inoculated with bacteria. All four samples were tested on the same rabbits to minimize the systemic errors. In order to avoid the influence of bacteria, we first evaluated bare catheters and HS/DAC coated catheters without bacteria. Four sets of parallel samples were tested, which means a total of four rabbits were used. The corresponding descriptions were corrected in revised manuscript and also Supporting information.

4. The authors have not fixed the error indicated by reviewer #2 (comment 2d). They still refer to 75% ethanol for sterilization on line 229 of the supporting information.

Our response: Thank you very much for your seriousness and responsibility, and we apologize for our negligence. The sentence was corrected as “Next, bare and HS/DAC-coated indwelling catheters were disinfected by immersing in 75% ethanol for 30 min” in revised Supporting information.

5. Figure 3b and supplementary figure 12(b) the vertical axis is labeled “IIa (arb. U at 405 nm).” I think that this is supposed to be “FXIIa (arb. U at 405 nm)”

Our response: Thank you for your suggestion. The vertical axis of Figure 3a,b and supplementary figure 12a,b was labeled as “FXa (arb. U at 405 nm)” and “FIIa (arb. U at 405 nm)” in the revised manuscript.

6. In the caption for figure 4b, the images are identified as from a coculture. Is this a coculture of two different cell types, or were the E. coli and S. aureus added cultured

on separate samples? I think coculture is the incorrect term to describe these experiments, but perhaps I do not understand the experiment correctly.

Our response: We feel very sorry for this confused description! The sentence was modified as “b SEM and CLSM images of bacteria on the surface of bare TPU and HS/DAC-coated TPU films after culture with *S. aureus* or *E. coli* for 24 h at 37 °C”.

7. Page 16, line 321 mentions “these four groups.” In the preceding text describing the results of this experiment there is no description of what the four groups are. These four groups should be briefly described for the reader here, since the methods are included in the paper after the results section. This is just one example of how the results are difficult to interpret. More context and rationale for each experiment (e.g., exactly what hypothesis is being tested in each experiment, why were the conditions used chosen, and how should the outcome of each experiment be interpreted in light of the other experiments) should be included in the results section.

Our response: Reviewer is right! (1) The sentence was modified as “Moreover, the internal thrombi of bare catheter, bare catheter with bacteria, coated catheter and coated catheter with bacteria were studied by SEM (Fig. 5d iv)” in the revised manuscript. Other similar issues in the article have also been revised. For example, the sentences were modified as “Owing to the strong hydrophobic interactions, DAC with alkyl chain length of 18 could exhibit more water aggregates than the other groups (alkyl chain < 18) at the same concentration of 4×10^{-5} mol/mL, and this HS/DAC complex was chosen for the following characterizations.”; “We further investigated the long-term stability of HS/DAC coating in flowing 0.9% NaCl and *in vivo* using a multi-channel circulation model and a subcutaneous implantation model, respectively”; “To further demonstrate the delayed clotting of HS/DAC-coated catheters, an *ex vivo* circuit model experiment was carried out, and the bare catheters and HS/DAC-coated catheters were implanted in rabbits via an arteriovenous shunt model (Supplementary Fig. 10a)”; “Compared with commercial bare CVC, the HS/DAC-coated CVC reduced the mass of blood clots by 84.6% (Supplementary Fig. 11b)”; “Therefore, to explore the salt-triggered effect of the HS/DAC coating, we examined the physicochemical properties of the coating before and after treatment in salt solution” in the revised manuscript.

(2) The Results section was rewritten according to the reviewer's suggestion. We add the background and purpose of each experiment at the beginning of the description. For example, “The formation of CRT is a multi-component pathology after catheter

implantation into blood vessels, such as various coagulation factors, proteins, blood cells, and inflammatory cells, etc. Therefore, the coagulation mechanism of the coating was firstly investigated by exploring the relationship between the surface and the coagulation cascade, protein, and inflammatory cells.”; “The conformation of antithrombin III changes after binding to HS, which can inhibit the activation of coagulation factor Xa and thrombin (IIa), thereby preventing the occurrence of coagulation. To examine the anticoagulant activity of the HA/DAC coating, chromogenic anti-FXa and anti-FIIa assays were used.” “Fibrinogen (Fg) activation to fibrin is the final step of the coagulation cascade, in which the hydrolysis of thrombin plays a crucial role. HS/DAC coating has been proven to have a high inhibitory effect on thrombin activity. Then, the adhesion/activation of Fg were further investigated through a closed circular tube loop of whole blood (Supplementary Fig. 8).”; “Conformational changes in Fg structure during Fg activation to form a fibrin network result in exposure of multiple binding sites. These binding sites can bind to various proteins, causing protein adsorption. Hence, the effect of HS/DAC coating on Fg activation after 2 h of circulation in whole blood was investigated by assessing the exposure of the γ chain via the immunofluorescence staining method.”; “Adherent Fg or platelets can promote the adhesion of leukocyte. Leukocyte, in turn, enhance the activation of local platelets and participate in the formation of thrombus. Therefore, monocytes and lymphocytes, as typical inflammatory cells, were selected to explore the leukocyte adhesion.”; “Anticoagulant coatings have important significance and application in various blood-contact medical devices. Firstly, the anticoagulant function of HS/DAC coating was explored by clotting time *in vitro*.”; “To further demonstrate the delayed clotting of HS/DAC-coated catheters, an *ex vivo* circuit model was carried out,” “Then beagle canine more consistent with acute thrombosis were selected as animal models to illustrate the antithrombotic properties of HS/DAC-coated commercial CVCs *in vivo*”; “The long-lasting antithrombotic stability of HS/DAC coating is another crucial for the implantation of medical catheters. Therefore, the antithrombotic stability HS/DAC-coated TPU and its HS/QAC control was initially investigated via chromogenic assays of anti-FXa and anti-FIIa after incubated in 0.9% NaCl solution of 30 days.”; “The hydrodynamic behavior in the human body is also a challenge to the stability of the coating. Therefore, we further investigated the long-term antithrombotic stability of HS/DAC coatings in artificial blood with shear rates of 30 s^{-1} and 300 s^{-1} (Supplementary Fig. 13a).”; “Finally, a rat subcutaneous implantation model was established to verify the antithrombotic

stability of HS/DAC coating *in vivo*.”

8. What species and cell type was used for the cell viability studies, and what was the rationale used for this selection?

Our response: L929 mouse fibroblast cells were used for the cell viability studies. L929 cells were derived from male C3H/An mice at 100 days. The first cell line recommended by the National standard (GB/T 16886) for biotoxicity evaluation of medical devices is the L929 mouse fibroblast cell, so this cell line was selected for testing. A large number of literatures has also used L929 cells to explore cytotoxicity (Nat Commun., 2022,13, 533; Mater. Today.,2021, 44, 25; Adv. Funct. Mater., 2021, 31, 2011165;).

9. The sentence beginning “DAC ...” on lines 95-97 is grammatically incorrect and needs to be clarified.

Our response: Thanks to the reviewer for your seriousness. The sentence was corrected as “Owing to the strong hydrophobic interactions, DAC with alkyl chain length of 18 could exhibit more water aggregates than the other groups (alkyl chain < 18) at the same concentration of 4×10^{-5} mol/mL, and this HS/DAC complex was chosen for the following characterizations.” in revised manuscript.

REVIEWER COMMENTS

Reviewer #3 (Remarks to the Author):

Review to Ms 2023-03 Nat Commun 22-43549C

Lin Liu, et al., Heparin-Network-Mediated Long-lasting Coatings on Intravascular Catheters for Adaptive Antithrombosis and Antibacterial Infection

Although the authors made many changes in the text, many comments of the reviewer are not addressed appropriately. The reviewer does not recommend the publication in this form.

Line 171: The manuscript still states that AT III with heparin inhibits the activation of FXa and thrombin. This phrasing is not correct: AT III inhibits the active enzymes, not their activation step.

Line 174: The phrasing “reduced the absorbance of FXa and FIIa” is not precise. “...reduced the substrate conversion...” would be better.

Line 184f: The text still presents two highly different experimental settings within one sentence without clarification: Fg adsorption and activation were determined from whole blood (which contains many competing proteins), whereas albumin adsorption was determined from a buffered solution of only albumin. As requested by the reviewer before, this difference needs to be expressed in the text and in the figure caption.

Figure 3c: The axis title still is not informative. It should express the parameter BSA adsorption. In the rebuttal letter, the authors state about modification of the caption to figure 3, however, these are not present in the provided version of the manuscript.

Line 217f: The “Beagle canine animal model” needs more description in the text, i.e. catheter implantation into the jugular veins needs to be mentioned.

For the figure R1, the authors tested the heparin release using elemental analysis. Bioactivity tests, preferably anti-FXa activity or an APTT assay with a mixture of extract and plasma would be more sensitive.

The authors clarify the abbreviation HS for heparin sodium (sodium salt of heparin) in the text. However, this term is misleading (or even wrong), as the coating described here is based on the salt formation of heparin with DAC, i.e. sodium is replaced by DAC.

The reviewer cannot follow the author’s argumentation against (neutrophil) granulocyte analysis on the surface. CD15 antibodies are widely available with all types of fluorescent dyes from multiple suppliers. These cells also have a characteristic cell nucleus, which is easily identified in routine DAPI staining.

Figure 4i still contains the term “underflow”

Reviewer #4 (Remarks to the Author):

The responses are mostly sufficient. A few errors remain, some introduced by the new edits to the manuscript. If the comments below are addressed, I would recommend this for publication:

1. In response to my previous comment number 3 the authors have provided some satisfactory response. All of this information should be added to the revised manuscript. To be clear, I was not requesting this information for my own satisfaction, but because clearly reporting the number of animals used in each study is essential. For example, the only place in the manuscript where the number of dogs used is referenced is in the caption of a supplementary figure. Reporting in the figure caption is good, but this is essential information that should also be included where the experiments and results are discussed and interpreted in the text. Table 2 provided in the responses is useful and should be provided for readers somewhere in the manuscript (or supporting information).

2. This should be thoroughly edited by a native English speaker. Some language errors include:
Line 53 – is “hampering” the correct term here? Does antibiotic and iv heparin administration hamper or lead to these complications?
Line 74 – “absorb” or “adsorb”?
Line 193 – change “leukocyte” to “leukocytes” (two occurrences).
Line 397 – change “significant” to “significantly”
Line 450 – change “underflow” to “under flow”

Point-to-point response:

Reviewer #3:

Although the authors made many changes in the text, many comments of the reviewer are not addressed appropriately. The reviewer does not recommend the publication in this form.

Our response: We greatly appreciate for the reviewer's professional comments and responsibility.

1. Line 171: The manuscript still states that AT III with heparin inhibits the activation of FXa and thrombin. This phrasing is not correct: AT III inhibits the active enzymes, not their activation step.

Our response: Reviewer is right! We have revised the inappropriate expression as follows:

“The conformation of antithrombin III changes after binding to HS, which can greatly inhibit the coagulation factor Xa (FXa) and thrombin (FIIa).” in line 170-172.

2. Line 174: The phrasing “reduced the absorbance of FXa and FIIa” is not precise. “...reduced the substrate conversion...” would be better.

Our response: We have revised the inappropriate expression as follows:

“Compared with TPU and DAC-TPU, (HS/DAC)-TPU reduced the substrate conversion by 95.8% and 94.3%, respectively (Fig. 3a, b), indicating that (HS/DAC)-TPU significantly inhibited FXa and FIIa” in line 173-175.

3. Line 184f: The text still presents two highly different experimental settings within one sentence without clarification: Fg adsorption and activation were determined from whole blood (which contains many competing proteins), whereas albumin adsorption was determined from a buffered solution of only albumin. As requested by the reviewer before, this difference needs to be expressed in the text and in the figure caption.

Our response: Thank you again! We have revised these inappropriate expressions of two experimental settings in the text as follows:

“Then, the adhesion/activation of Fg were further investigated through a closed circular tube loop of whole blood (Supplementary Fig. 8). FITC-Fg was added to fresh blood and cycled with samples for 2 h to study Fg adhesion. Confocal laser scanning microscopy (CLSM) images indicated that Fg was less adhered in the HS/DAC-coated catheter than the bare catheter (Fig. 3d). On the contrary, negligible differences were

observed in the amount of bovine serum albumin (BSA) by bicinchoninic acid (BCA) assay when samples soaked in PBS solution of BSA (Fig. 3c)” in line 179-185.

The difference of experimental settings was also revised in the caption of Figure 3 as follows:

“c Amounts of BSA absorbed on bare and HS/DAC-coated TPU. The sample was immersed in BSA solution and the adsorption of proteins was determined by a BCA protein assay kit (n = 9). d CLSM images of Fg adhesion, activation and leukocyte adhesion on the surface of bare catheters and HS/DAC-coated catheters (n = 3). FITC-Fg was added to fresh whole blood and cycled with samples through a closed circular tube loop for 2 h in the assay of Fg adhesion and activation. Leukocyte adhesion was detected via immunofluorescent” in line 248-253.

4. Figure 3c: The axis title still is not informative. It should express the parameter BSA adsorption. In the rebuttal letter, the authors state about modification of the caption to figure 3, however, these are not present in the provided version of the manuscript.

Our response: We feel very sorry for this carelessness! The axis title of Figure 3c was modified as “OD_{562 nm} qualitatively BSA adhesion”. The modification was also expressed in the caption of Figure 3 as follows: “c Amounts of BSA absorbed on bare and HS/DAC-coated TPU. The sample was immersed in BSA solution and the adsorption of proteins was determined by a BCA protein assay kit (n = 9)” in line 249-251.

Fig. 3c Amounts of BSA absorbed on bare and HS/DAC-coated TPU. The sample was immersed in BSA solution and the adsorption of proteins was determined by a BCA protein assay kit (n = 9).

5. Line 217f: The “Beagle canine animal model” needs more description in the text, i.e.

catheter implantation into the jugular veins needs to be mentioned.

Our response: More descriptions for this “Beagle canine animal model” were added in the revised manuscript and also given as follows:

“Commercial CVC and HS/DAC-coated CVC were aseptically inserted into the left and right jugular veins of canines, respectively. After 24 h of implantation, the jugular vein was removed and opened lengthwise” in line 218-220.

6. For the figure R1, the authors tested the heparin release using elemental analysis. Bioactivity tests, preferably anti-FXa activity or an APTT assay with a mixture of extract and plasma would be more sensitive.

Our response: Thank you for your advices! We have investigated the release of heparin by chromogenic anti-FXa assay and anti-FIIa assay. “The long-lasting antithrombotic stability of HS/DAC coating is another crucial for the implantation of medical catheters. Therefore, the antithrombotic stability HS/DAC-coated TPU and its HS/QAC control was initially investigated via chromogenic assays of anti-FXa and anti-FIIa after incubated in 0.9% NaCl solution of 30 days. Contributing to the strong covalent bonding of HS/DAC coating, the inhibitory activity against FXa and FIIa was maintained well even after one month (Supplementary Fig. 12).” at line 225-230 in manuscript. “HS/QAC-coated TPU and HS/DAC-coated TPU were immersed into 0.9% NaCl solution with sharking (100 rpm) at 37 °C. Samples were removed on different days (0, 10, 20, 30 days) and washed with DI water. After that, the anticoagulant ability of the samples was determined by chromogenic anti-FXa assay and anti-FIIa assay described above.” at line 156-160 in Supporting Information.

Supplementary Fig. 12 Anti-FXa (a) and anti-FIIa (b) assays of the coating after incubation in normal saline for 0, 10, 20 and 30 days (n = 3).

7. The authors clarify the abbreviation HS for heparin sodium (sodium salt of heparin) in the text. However, this term is misleading (or even wrong), as the coating described here is based on the salt formation of heparin with DAC, i.e. sodium is replaced by DAC.

Our response: To distinguish the confused description between reactant and product, we revised the abbreviation of heparin sodium as HS-Na and the surfactant dimethyloctadecyl [3-(trimethoxysilyl)propyl]ammonium chloride as DAC-Cl. The complex formed by HS-Na and DAC-Cl is labeled HS/DAC. These modifications were revised at line 72-73, 134-136 in the revised manuscript and at line 24-25, 31-32, 40-44, 49-50 and Supplementary Fig. 2 and 4 in the revised Supporting Information.

8. The reviewer cannot follow the author's argumentation against (neutrophil) granulocyte analysis on the surface. CD15 antibodies are widely available with all types of fluorescent dyes from multiple suppliers. These cells also have a characteristic cell nucleus, which is easily identified in routine DAPI staining.

Our response: We agree with reviewer's suggestions to study the granulocyte analysis on the surface by using fluorescent-labeled CD15 antibodies and DAPI staining. Considering the fact that the species reactivity of fluorescent-labeled CD15 antibodies is predominantly human, it is difficult to conduct this antibody to qualify our unit from rats. Hence, the nuclei of neutrophils were labeled by using the DAPI staining method. The typical lobulated neutrophil nuclei were selected for analysis by CLSM, indicating the less neutrophil adhesion in our coated catheter than bare catheter. And the neutrophils adhesion assay was added in revised manuscript and also listed in the following.

"Therefore, monocytes, lymphocytes and neutrophils, as typical inflammatory cells, were selected to explore the leukocytes adhesion. Benefiting to the strong ability of HS/DAC coatings to reduce Fg adhesion and activation, a large number of inflammatory cells passively trapped in fibrin clots were observed only on the surface of the bare catheter (Fig. 3d and Supplementary Fig. 8b)." at line 193-197.

"For the neutrophil adhesion test, DAPI with 405 nm excitation was used to label the cell nucleus. Catheter segments were fixed in Triton X-100 (diluted 1:1000 in PBS) for 10 min and cleaned with PBS for three times. Next, samples were immersed in DAPI (10 µg/mL) solution for 10 min and rinsed with PBS for one time. Finally, samples were observed by CLSM." at line 122-126 and Supplementary Fig. 8b in the revised Supporting Information.

Supplementary Fig. 8 (b) CLSM images of neutrophils adhesion on the surface of bare catheters and HS/DAC-coated catheters (n = 3).

9. Figure 4i still contains the term “underflow”

Our response: We have revised the “underflow” as “Flow incubation” in Figure 4i as follows:

Fig. 4i The long-term antibacterial stability of HS/DAC coating in artificial blood with shear rates of 0, 30, and 300 s⁻¹ after 30 days of flow (n = 4).

Thank you again!

Reviewer #4:

The responses are mostly sufficient. A few errors remain, some introduced by the new edits to the manuscript. If the comments below are addressed, I would recommend this for publication:

Our response: Thank you very much for your seriousness, and we apologize for our carelessness.

1. In response to my previous comment number 3 the authors have provided some satisfactory response. All of this information should be added to the revised manuscript. To be clear, I was not requesting this information for my own satisfaction, but because clearly reporting the number of animals used in each study is essential. For example, the only place in the manuscript where the number of dogs used is referenced is in the caption of a supplementary figure. Reporting in the figure caption is good, but this is essential information that should also be included where the experiments and results are discussed and interpreted in the text. Table 2 provided in the responses is useful and should be provided for readers somewhere in the manuscript (or supporting information).

Our response: Reviewer is right! We have added the number of animals in experiments and results as follows:

“the bare catheters and HS/DAC-coated catheters were implanted in three rabbits via an arteriovenous shunt model (Supplementary Fig. 10a)” in line 211-212; “Commercial CVC and HS/DAC-coated CVC were aseptically inserted into the left and right jugular veins of three canines, respectively.” in line 218-219; “An indwelling catheter (24 G, TPU) inoculated with *S. aureus* was implanted into two sides of the mouse back (n = 9).” in line 295-296; “To further illustrate the biocompatibility in vivo, the histocompatibility of bare and HS/QAC-coated catheters was investigated by a subcutaneous implantation model in mice (n = 6) and a vasculature model in acute canine (n = 3).” in line 324-326; “The amount of thrombus on the bare catheter was much higher than that on the HS/DAC-coated catheter in four rabbits, especially on the bare catheter inoculated with bacteria (Fig. 5b).” in line 353-355. The contents of Table 2 are listed in detail in Reporting summary.

2. This should be thoroughly edited by a native English speaker. Some language errors include:

Line 53 – is “hampering” the correct term here? Does antibiotic and iv heparin

administration hamper or lead to these complications?

Line 74 – “absorb” or “adsorb”?

Line 193 – change “leukocyte” to “leukocytes” (two occurrences).

Line 397 – change “significant” to “significantly”

Line 450 – change “underflow” to “under flow”

Our response:

(1) We have revised the misleading expressions in new manuscript as follows: “Normally, a common strategy to treat infection and thrombi in the clinic is to administer antibiotics and intravenous heparin (HS) prior to or immediately following catheter implantations. However, the generation of antibiotic-resistant bacteria and heparin-caused thrombocytopenia are still challenges for these treatments.” in line 51-54.

(2) “absorb” was revised as “adsorb” in line 74 and line 252.

(3) “leukocyte” was revised as “leukocytes” in line 195.

(4) “significant” was revised as “significantly” in line 400.

(5) “underflow” was revised as “under flow incubation” in line 453.

Thank you again!

REVIEWERS' COMMENTS

Reviewer #3 (Remarks to the Author):

Review to Ms NCOMMS-22-43549D

Lin Liu, et al., Heparin-Network-Mediated Long-lasting Coatings on Intravascular Catheters for Adaptive Antithrombosis and Antibacterial Infection

The authors responded to all comments of the Reviewer appropriately. The text can be published in the present form.